# On the Size and Approximation Error of Distilled Sets

**Alaa Maalouf**[*]
MIT CSAIL

**Murad Tukan**[*]
DataHeroes

**Noel Loo**
MIT CSAIL

**Ramin Hasani**
MIT CSAIL

**Mathias Lechner**
MIT CSAIL

**Daniela Rus**
MIT CSAIL

## Abstract

Dataset Distillation is the task of synthesizing small datasets from large ones while still retaining comparable predictive accuracy to the original uncompressed dataset. Despite significant empirical progress in recent years, there is little understanding of the theoretical limitations/guarantees of dataset distillation, specifically, what excess risk is achieved by distillation compared to the original dataset, and how large are distilled datasets? In this work, we take a theoretical view on kernel ridge regression (KRR) based methods of dataset distillation such as Kernel Inducing Points. By transforming ridge regression in random Fourier features (RFF) space, we provide the first proof of the existence of small (size) distilled datasets and their corresponding excess risk for shift-invariant kernels. We prove that a small set of instances exists in the original input space such that its solution in the RFF space coincides with the solution of the original data. We further show that a KRR solution can be generated using this distilled set of instances which gives an approximation towards the KRR solution optimized on the full input data. The size of this set is linear in the dimension of the RFF space of the input set or alternatively near linear in the number of effective degrees of freedom, which is a function of the kernel, number of datapoints, and the regularization parameter $\lambda$. The error bound of this distilled set is also a function of $\lambda$. We verify our bounds analytically and empirically.

## 1 Introduction

Motivated by the growing data demands of modern deep learning, dataset distillation [ZB21, NNXL21, ZNB22, WZTE18] aims to summarize large datasets into significantly smaller synthetic *distilled* datasets, which when trained on retain high predictive accuracy, comparable to the original dataset. These distilled datasets have applications in continual learning [ZNB22, SCCB22], architecture search [SRL+19], and privacy preservation [CKF22]. Recent years have seen the development of numerous distillation algorithms, but despite this progress, the field has remained largely empirical. Specifically, there is little understanding of what makes one dataset "easier to distill" than another, or whether such small synthetic datasets even exist.

This work aims to fill this gap by providing the first bounds on the sufficient size and relative error associated with distilled datasets. Noting prior work relating neural network training to kernel ridge regression (KRR), we consider dataset distillation in the kernel ridge regression settings with shift-invariant kernels. By casting the problem into the Random Fourier Feature (RFF) space, we show that: **The size and relative error of distilled datasets is governed by the kernel's "number of effective degrees of freedom",** $d_k^\lambda$. Specifically, in Section 4, we show that distilled sets of size $\Omega(d_k^\lambda \log d_k^\lambda)$, exist, with $12\lambda + 2\mathcal{L}_\lambda$ predictive error on the training dataset, and only $8\lambda$ error with

---

[*]Equal contribution. Correspondence E-mail: alaam@mit.edu.

37th Conference on Neural Information Processing Systems (NeurIPS 2023).

respect to the optimal solution computed on the full dataset, where $\lambda$ is the kernel ridge regression regularization parameter and $\mathcal{L}_\lambda$ the KRR training error on the original dataset; see Theorem 3 and Remark 7 for full details.

**These bounds hold in practice for both real and synthetic datasets**. In section 5, we validate our theorem by distilling synthetic and real datasets with varying sizes and values of $d_k^\lambda$, showing that in all scenarios our bounds accurately predict the error associated with distillation.

## 2  Related work

**Coresets.**  Coresets are weighted selections from a larger training dataset, which, when used for training, yield similar outcomes as if the whole dataset was used [MSSW18, MBL20, PDM22, MEM+22]. The key benefit of using coresets is that they significantly speed up the training process, unlike when the full data set is used. Current coresets methods incorporate clustering techniques [FL11, LBK16, BLHK16], bilevel optimization [BMK20], sensitivity analysis [MSSW18, HCB16, TMF20, MSF20], and surrogate models for approximation [TZM+23]. Newer strategies are specifically designed for neural networks, where before each training epoch, coresets are chosen such that their gradients align with the gradients of the entire dataset [MBL20, PDM22, TZM+23], followed by training the model on the chosen coreset. Although coresets are usually theoretically supported, these methods fall short when the aim is to compute a coreset once for a full training procedure.

**Dataset Distillation.**  To this end, dataset distillation algorithms construct synthetic datasets (not necessarily a subset from the original input) such that gradient descent training on the synthetic datapoints results in high predictive accuracy on the real dataset. Cast as a bilevel optimization problem, early methods involve unrolling training computation graph [WZTE18] for a few gradient descent steps and randomly sampled weight initializations. More sophisticated methods aim to approximate the unrolled computation using kernel methods [NCL21, NNXL21, ZNB22, LHAR22a, LHLR23a], surrogate objectives such gradient matching [ZMB21, ZB21], trajectory matching [CWT+22] or implicit gradients [LHLR23b]. The kernel-induced points (KIP) algorithm [NCL21, NNXL21] is a technique that employs Neural Tangent Kernel (NTK) theory[JGH18, LHAR22b] to formulate the ensuing loss: $\mathcal{L}_{KIP} = \frac{1}{2}\|y_t - K_{TS}K_{SS}^{-1}y_S\|_2^2$. This loss signifies the predictive loss of training infinitely wide networks on distilled datapoints $X_S$ with corresponding labels $y_S$, on the original training set and labels $X_T, y_T$, with $K_{\cdot,\cdot}$ being the NTK. Dataset distillation is closely related to the use of inducing points to accelerate Gaussian Processes [SG05, TRB16], for which convergence rates exist, but the existence of such inducing points is not unknown [BRVDW19].

**From dataset distillation to kernel ridge regression.**  Kernel ridge regression (KRR) extends the linear machine learning ridge regression model by using a kernel function to map input data into higher-dimensional feature spaces, allowing for more complex non-linear relationships between variables to be captured [Mur12]. Various methods have been proposed to improve and accelerate the training process of kernel ridge regression. Most notably, Random Fourier Features [RR07] approximates shift-invariant kernel functions by mapping the input data into a lower-dimensional feature space using a randomized cosine transformation. This has been shown to work effectively in practice due to regularizing effects [JSS+20], as well as providing approximation bounds to the full kernel ridge regression [SS15, AKM+17, LTOS19]. Training infinite-width neural networks can be cast as kernel ridge regression with the Neural Tangent Kernel (NTK) [JGH18], which allows a closed-form solution of the infinite-width neural network's predictions, enabling kernel-based dataset distillation algorithms such as [NCL21, NNXL21, LHAR22a].

## 3  Background

**Goal.** We provide the first provable guarantees on the existence and approximation error of a small distilled dataset in the KRR settings. We first provide notations that will be used throughout the paper.

**Notations.** Let $\mathcal{H}$ be a Hilbert space with $\|\cdot\|_{\mathcal{H}}$ as its norm. For a vector $a \in \mathbb{R}^n$, we use $\|a\|_2$ to denote its Euclidean norm, and $a_i$ to denote its $i$th entry for every $i \in [n]$. For any positive integer $n$, we use the convention $[n] = \{1, 2, \cdots, n\}$. Let $A \in \mathbb{R}^{n \times m}$ be a matrix, then, for every $i \in [n]$ and $j \in [m]$, $A_{i*}$ denotes the $i$th row of $A$, $A_{*j}$ denotes the $j$th column of $A$, and $A_{i,j}$ is the $j$th entry of the $i$th row of $A$. Let $B \in \mathbb{R}^{n \times n}$, then we denote the trace of $B$ by $Tr(B)$. We use $\mathbf{I}_m \in \mathbb{R}^{m \times m}$ to denote the identity matrix. Finally, vectors are addressed as column vectors unless stated otherwise.

## 3.1 Kernel ridge regression

Let $\mathbf{X} \in \mathbb{R}^{n \times d}$ be a matrix and let $y \in \mathbb{R}^n$ be a vector. Let $k : \mathbb{R}^d \times \mathbb{R}^d \to [0, \infty)$ be a kernel function, and let $\mathbf{K} \in \mathbb{R}^{n \times n}$ be its corresponding kernel matrix with respect to the rows of $\mathbf{X}$; i.e., $\mathbf{K}_{i,j} = k(\mathbf{X}_{i*}, \mathbf{X}_{j*})$ for every $i, j \in [n]$. Let $\lambda > 0$ be a regularization parameter. The goal of kernel ridge regression (KRR) involving $\mathbf{X}, y, k$, and $\lambda$ is to find

$$\alpha_{[\mathbf{X},y,k]}^{\lambda} \in \arg\min_{\alpha \in \mathbb{R}^n} \frac{1}{n} \|y - \mathbf{K}\alpha\|_2^2 + \lambda\alpha^T \mathbf{K}\alpha. \tag{1}$$

We use the notation $f_{[\mathbf{X},y,k]}^{\lambda} : \mathbb{R}^d \to \mathbb{R}$ to denote the in-sample prediction by applying the KRR solution obtained on $\mathbf{X}, y$ and $\lambda$ using the kernel $k$, i.e., for every $x \in \mathbb{R}^d$,

$$f_{[\mathbf{X},y,k]}^{\lambda}(x) = \sum_{i=1}^{n} \alpha_{[\mathbf{X},y,k]_i}^{\lambda} k(\mathbf{X}_{i*}, x). \tag{2}$$

To provide our theoretical guarantees on the size and approximation error for the distilled datasets, the following assumption will be used in our theorem and proofs.

**Assumption 1.** *We inherit the same theoretical assumptions used at [LTOS21] for handling the KRR problem: (I) Let $\mathcal{F}$ be the set of all functions mapping $\mathbb{R}^d$ to $\mathbb{R}$. Let $f^* \in \mathcal{F}$ be the minimizer of $\sum_{i=1}^{n} |y_i - f(\mathbf{X}_{i*})|^2$, subject to the constraint that for every $x \in \mathbb{R}^d$ and $y \in \mathbb{R}$, $y = f^*(x) + \epsilon$, where $\mathbb{E}(\epsilon) = 0$ and Var$(\epsilon) = \sigma^2$. Furthermore, we assume that $y$ is bounded, i.e., $|y| \le y_0$. (II) We assume that $\left\| f_{[\mathbf{X},y,k]}^{\lambda} \right\|_{\mathcal{H}} \le 1$. (III) For a kernel $k$, denote with $\lambda_1 \ge \cdots \ge \lambda_n$ the eigenvalues of the kernel matrix $\mathbf{K}$. We assume that the regularization parameter satisfies $0 \le n\lambda \le \lambda_1$.*

**The logic behind our assumptions.** First, the idea behind Assumption (I) is that the pair $(\mathbf{X}, y)$ can be linked through some function that can be from either the same family of kernels that we support (i.e., shift-invariant) or any other kernel function. In the context of neural networks, the intuition behind Assumption (I) is that there exists a network from the desired architectures that gives a good approximation for the data. Assumption (II) aims to simplify the bounds used throughout the paper as it is a pretty standard assumption, characteristic to the analysis of random Fourier features [LTOS19, RR17]. Finally, Assumption (III) is to prevent underfitting. Specifically speaking, the largest eigenvalue of $\mathbf{K}(\mathbf{K} + n\lambda\mathbf{I}_n)^{-1}$ is $\frac{\lambda_1}{(\lambda_1 + n\lambda)}$. Thus, in the case of $n\lambda > \lambda_1$, the in-sample prediction is dominated by the term $n\lambda$. Throughout the following analysis, we will use the above assumptions. Hence, for the sake of clarity, we will not repeat them, unless problem-specific clarifications are required.

**Connection to Dataset distillation of neural networks.** Since the neural network kernel in the case of infinite width networks describes a Gaussian distribution [JGH18], we aim at proving the existence of small sketches (distilled sets) for the input data with respect to the KRR problem with Gaussian kernel. However, the problem with this approach is that the feature space (in the Gaussian kernel corresponding mapping) is rather intangible or hard to map to, and sketch (distilled set) construction techniques require the representation of these points in the feature space.

To resolve this problem, we use a randomized approximated feature map, e.g., random Fourier features (RFF), and weighted random Fourier features (Weighted RFF). The dot product between every two mapped vectors in this approximated feature map aims to approximate their Gaussian kernel function [RR07]. We now restate a result connecting ridge regression in the RFF space (or alternatively weighted RFF), and KRR in the input space.

**Theorem 2** (A result of the proof of Theorem 1 and Corollary 2 of [LTOS21])**.** *Let $\mathbf{X} \in \mathbb{R}^{n \times d}$ be an input matrix, $y \in \mathbb{R}^n$ be an input label vector, $k : \mathbb{R}^d \times \mathbb{R}^d \to [0, \infty)$ be a shift-invariant kernel function, and $\mathbf{K} \in \mathbb{R}^{n \times n}$, where $\forall i, j \in [n] : \mathbf{K}_{i,j} = k(\mathbf{X}_{i*}, \mathbf{X}_{j*})$. Let $\lambda > 0$, and let $d_{\mathbf{K}}^{\lambda} = Tr\left(\mathbf{K}(\mathbf{K} + n\lambda\mathbf{I}_n)^{-1}\right)$. Let $s_\phi \in \Omega\left(d_{\mathbf{K}}^{\lambda} \log\left(d_{\mathbf{K}}^{\lambda}\right)\right)$ be a positive integer. Then, there exists a pair $(\phi, \widetilde{\mathbf{X}})$ such that (i) $\phi$ is a mapping $\phi : \mathbb{R}^d \to \mathbb{R}^{s_\phi}$ (which is based on either the weighted RFF function or the RFF function [LTOS21]), (ii) $\widetilde{\mathbf{X}}$ is a matrix $\widetilde{\mathbf{X}} \in \mathbb{R}^{n \times s_\phi}$ where for every $i \in [n]$,*

$\widetilde{\mathbf{X}}_{i*} := \phi\left(\mathbf{X}_{i*}\right)$, and (iii) $(\phi, \widetilde{\mathbf{X}})$ satisfies

$$\frac{1}{n}\sum_{i=1}^{n}\left|y_i - f^{\lambda}_{[\widetilde{\mathbf{X}},y,\phi]}\left(\widetilde{\mathbf{X}}_{i*}\right)\right|^2 \leq \frac{1}{n}\sum_{i=1}^{n}\left|y_i - f^{\lambda}_{[\mathbf{X},y,k]}\left(\mathbf{X}_{i*}\right)\right|^2 + 4\lambda,$$

where $f^{\lambda}_{[\widetilde{\mathbf{X}},y,\phi]}$ : $\mathbb{R}^{s_\phi} \to \mathbb{R}$ such that for every row vector $z \in \mathbb{R}^{s_\phi}$, $f^{\lambda}_{[\widetilde{\mathbf{X}},y,\phi]}(z) = z\left(\widetilde{\mathbf{X}}^T\widetilde{\mathbf{X}} + \lambda n s_\phi \lambda \mathbf{I}_{s_\phi}\right)^{-1}\widetilde{\mathbf{X}}^T y$. Note that, Table 1 gives bounds on $s_\phi$ when $\lambda \propto \frac{1}{\sqrt{n}}$.

**Intuition behind Theorem 2.** Theorem 2 bounds the difference (additive approximation error) between (i) the MSE loss between the ground truth labels and the predictions obtained by applying Kernel Ridge regression (KRR) on the raw (original) data, and (ii) the MSE between the ground truth labels and the predictions obtained when applying Ridge regression on the mapped (full) training data via random Fourier features (*RFF*). Theorem 2 will be utilized to set the minimal dimension of the *RFF* which yields the desired additive approximation, i.e., $4\lambda$. Thus the intuition behind using this theorem is to link the dimension of the *RFF* with the size of the distilled set. In other words, we use this error bound and sufficient size (of the minimal dimension of the *RFF*) to provide proof of the sufficient small size of the distilled set.

# 4   Main result: on the existence of small distilled sets

In what follows, we show that for any given matrix $\mathbf{X} \in \mathbb{R}^{n \times d}$ and a label vector $y \in \mathbb{R}^n$, there exists a matrix $\mathbf{S} \in \mathbb{R}^{s_\phi \times d}$ and a label vector $y_{\mathbf{S}} \in \mathbb{R}^r$ such that the fitting solution in the RFF space mapping of $\mathbf{S}$ is identical to that of the fitted solution on the RFF space mapping of $\mathbf{X}$. With such $\mathbf{S}$ and $y_{\mathbf{S}}$, we proceed to provide our main result showing that one can construct a solution for KRR in the original space of $\mathbf{S}$ which provably approximates the quality of the optimal KRR solution involving $\mathbf{X}$ and $y$. Thus, we obtain bounds on the minimal distilled set size required for computing a robust approximation, as well as bounds on the error for such a distilled set.

We now provide Theorem 3 followed by its proof of the existence of a small distilled set. Then we provide extensive details and intuitive explanations about the steps of the proof.

**Theorem 3** (On the existence of some distilled data). *Let* $\mathbf{X} \in \mathbb{R}^{n \times d}$ *be a matrix,* $y \in \mathbb{R}^n$ *be a label vector,* $k : \mathbb{R}^d \times \mathbb{R}^d \to [0, \infty)$ *be a kernel function,* $\Upsilon = (0,1) \cup \{2\}$, *and let* $s_\phi$ *and* $\widetilde{\mathbf{X}}$ *be defined as in Theorem 2. Assume that the rank of* $\widetilde{\mathbf{X}}$ *is* $s_\phi$, *then, there exists a matrix* $\mathbf{S} \in \mathbb{R}^{s_\phi \times d}$ *and a label vector* $y_{\mathbf{S}}$ *such that*

*(i) the weighted RFF mapping* $\widetilde{\mathbf{S}} \in \mathbb{R}^{s_\phi \times s_\phi}$ *of* $\mathbf{S}$, *satisfies that* $\left(\widetilde{\mathbf{X}}^T\widetilde{\mathbf{X}} + \lambda n s_\phi \lambda \mathbf{I}_{s_\phi}\right)^{-1}\widetilde{\mathbf{X}}^T y = \left(\widetilde{\mathbf{S}}^T\widetilde{\mathbf{S}} + \lambda n s_\phi \lambda \mathbf{I}_{s_\phi}\right)^{-1}\widetilde{\mathbf{S}}^T y_{\mathbf{S}}$, *and*

*(ii) there exists an in-sample prediction* $f^{\lambda,\mathbf{X},y}_{[\mathbf{S},y_{\mathbf{S}},k]}$ *(not necessarily the optimal on* $\mathbf{S}$ *and* $y_{\mathbf{s}}$*) satisfying*

$$\frac{1}{n}\sum_{i=1}^{n}\left|f^{\lambda}_{[\mathbf{X},y,k]}\left(\mathbf{X}_{i*}\right) - f^{\lambda,\mathbf{X},y}_{[\mathbf{S},y_{\mathbf{S}},k]}\left(\mathbf{X}_{i*}\right)\right|^2 \leq \min_{\tau \in \Upsilon}\left(2\max\left\{\tau, \frac{4}{\tau^2}\right\} + 2\min\left\{1+\tau, \frac{4(1+\tau)}{3\tau}\right\}\right)\lambda,\tag{3}$$

*and*

$$\frac{1}{n}\sum_{i=1}^{n}\left|y_i - f^{\lambda,\mathbf{X},y}_{[\mathbf{S},y_{\mathbf{S}},k]}\left(\mathbf{X}_{i*}\right)\right|^2 \leq \min_{\tau \in \Upsilon}\frac{\min\left\{1+\tau, \frac{4(1+\tau)}{3\tau}\right\}}{n}\sum_{i=1}^{n}\left|y_i - f^{\lambda}_{[\mathbf{X},y,k]}\left(\mathbf{X}_{i*}\right)\right|^2 + \left(4\min\left\{1+\tau, \frac{4(1+\tau)}{3\tau}\right\} + 2\max\left\{\tau, \frac{4}{\tau^2}\right\}\right)\lambda.\tag{4}$$

*Proof.* Let $\mathbf{S}$ be any matrix in $\mathbb{R}^{s_\phi \times d}$ such its weighted RFF mapping $\widetilde{\mathbf{S}}$ is of rank equal to that of $\widetilde{\mathbf{X}}$ and for every $i \in [s_\phi]$, $\sum_{j=1}^{n} k\left(\mathbf{S}_{i*}, \mathbf{X}_{i*}\right) \neq 0$.

**Proof of (i).** To ensure (i), we need to find a corresponding proper $y_{\mathbf{S}}$. We observe that

$$\left(\widetilde{\mathbf{S}}^T \widetilde{\mathbf{S}} + \lambda n s_\phi \lambda \mathbf{I}_{s_\phi}\right)\left(\widetilde{\mathbf{X}}^T \widetilde{\mathbf{X}} + \lambda n s_\phi \lambda \mathbf{I}_{s_\phi}\right)^{-1} \widetilde{\mathbf{X}}^T y = \widetilde{\mathbf{S}}^T y_{\mathbf{S}}$$

Let $b = \left(\widetilde{\mathbf{S}}^T \widetilde{\mathbf{S}} + \lambda n s_\phi \lambda \mathbf{I}_{s_\phi}\right)\left(\widetilde{\mathbf{X}}^T \widetilde{\mathbf{X}} + \lambda n s_\phi \lambda \mathbf{I}_{s_\phi}\right)^{-1} \widetilde{\mathbf{X}}^T y$, be the left-hand side term above. $b$ is a vector of dimension $s_\phi$. Hence we need to solve $b = \widetilde{\mathbf{S}}^T y_{\mathbf{S}}$ for $y_{\mathbf{S}}$. Since $\mathbf{S}$ has full rank then we have a linear system involving $s_\phi$ variables and $s_\phi$ equations. Thus, the solution to such a system exists and is $y_{\mathbf{S}} = \left(\widetilde{\mathbf{S}}^T\right)^\dagger b$, where $(\cdot)^\dagger$ denotes the pseudo-inverse of the given matrix.

**Proof of (ii).** Inspired by [LHAR22a] and [NCL20], the goal is to find a set of instances that their in-sample prediction with respect to the input data ($\mathbf{X}$ in our context) would lead to an approximation towards the solution that one would achieve if the KRR was used only with the input data. To that end, we introduce the following Lemm.

**Lemma 4** (Restatement of Lemma 6 [LTOS21]). *Under Assumption 1 and the definitions in Theorem 2, for every $f \in \mathcal{H}$ with $\|f\|_{\mathcal{H}} \leq 1$, with constant probability, it holds that*

$$\inf_{\substack{\sqrt{s_\phi}\|\beta\|_2 \leq \sqrt{2} \\ \beta \in \mathbb{R}^{s_\phi}}} \sum_{i=1}^{n} \frac{1}{n}\left|f\left(\mathbf{X}_{i*}\right) - \widetilde{\mathbf{X}}_{i*}\beta\right|^2 \leq 2\lambda.$$

Note that Lemma 4 shows that for every in-sample prediction function with respect to $\mathbf{X}$, there exists a query $\beta \in \mathbb{R}^{s_\phi}$ in the RFF space of that input data such that the distance between the in-prediction sample function in the input space and the in-sample prediction in the RFF space is at $2\lambda$. Furthermore, at [LTOS21] it was shown that $\beta$ is defined as $\beta = \frac{1}{s_\phi}\widetilde{\mathbf{X}}^T\left(\widetilde{\mathbf{X}}\widetilde{\mathbf{X}}^T + n\lambda\mathbf{I}_{s_\phi}\right)^{-1}\mathbf{f}[\mathbf{X}]$, where $\mathbf{f}[\mathbf{X}]_i = f\left(\mathbf{X}_{i*}\right)$ for every $i \in [n]$.

We thus set out to find an in-sample prediction function that is defined over $\mathbf{S}$ such that by its infimum by Lemma 4 would be the same solution $\beta$ that the ridge regression on $\widetilde{\mathbf{X}}$ attains with respect to the $y$. Specifically speaking, we want to find an in-sample prediction $f_{[\mathbf{S},y_{\mathbf{S}},k]}^{\lambda,\mathbf{X},y}(\cdot)$ such that

$$\beta = \frac{1}{s_\phi}\widetilde{\mathbf{X}}^T\left(\frac{1}{s_\phi}\widetilde{\mathbf{X}}\widetilde{\mathbf{X}}^T + n\lambda\mathbf{I}_{s_\phi}\right)^{-1}\mathbf{f}_{\mathbf{S}}[\mathbf{X}], \tag{5}$$

where (i) $\mathbf{f}_{\mathbf{S}}[\mathbf{X}] \in \mathbb{R}^n$ such that for every $i \in [n]$, $\mathbf{f}_{\mathbf{S}}[\mathbf{X}]_i = f_{[\mathbf{S},y_{\mathbf{S}},k]}^{\lambda,\mathbf{X},y}(\mathbf{X}_{i*})$, and (ii) $f_{[\mathbf{S},y_{\mathbf{S}},k]}^{\lambda,\mathbf{X},y}(\cdot) = \sum_{i=1}^{s_\phi+1} \alpha_i k\left(\mathbf{S}_{i*}, \cdot\right)$ such that $\alpha \in \mathbb{R}^{s_\phi}$.

Hence we need to find an in-sample prediction function $f_{[\mathbf{S},y_{\mathbf{S}},k]}^{\lambda,\mathbf{X},y}$ satisfying 5. Now, notice that $\beta \in \mathbb{R}^{s_\phi}$, $\mathbf{f}_{\mathbf{S}}[\mathbf{X}] \in \mathbb{R}^n$ and $\widetilde{\mathbf{X}}^T\left(\frac{1}{s_\phi}\widetilde{\mathbf{X}}\widetilde{\mathbf{X}}^T + n\lambda\mathbf{I}_n\right)^{-1} \in \mathbb{R}^{s_\phi \times n}$. Due to the fact that we aim to find $f_{[\mathbf{S},y_{\mathbf{S}},k]}^{\lambda,\mathbf{X},y}$, such a task boils down to finding $\alpha \in \mathbb{R}^{s_\phi}$ which defines $f_{[\mathbf{S},y_{\mathbf{S}},k]}^{\lambda,\mathbf{X},y}$ as in (ii). The above problem can be reduced to a system of linear equations where the number of equalities is $s_\phi$, while the number of variables is $s_\phi$.

To do so, we denote $\frac{1}{s_\phi}\widetilde{\mathbf{X}}^T\left(\frac{1}{s_\phi}\widetilde{\mathbf{X}}\widetilde{\mathbf{X}}^T + n\lambda\mathbf{I}_n\right)^{-1}$ by $\hat{\mathbf{A}}$, and observe that we aim to solve

$$\beta = \hat{\mathbf{A}}f_{\mathbf{S}}^\lambda[\mathbf{X}] = \hat{\mathbf{A}}\begin{bmatrix} \sum_{i=1}^{s_\phi} \alpha_i k\left(\mathbf{S}_{i*}, \mathbf{X}_{1*}\right) \\ \sum_{i=1}^{s_\phi} \alpha_i k\left(\mathbf{S}_{i*}, \mathbf{X}_{2*}\right) \\ \vdots \\ \sum_{i=1}^{s_\phi} \alpha_i k\left(\mathbf{S}_{i*}, \mathbf{X}_{n*}\right) \end{bmatrix}.$$

We now show that every entry $b_j$ ($j \in [s_\phi]$) in $\beta$ can be rewritten as inner products between another pair of vectors in $\mathbb{R}^{s_\phi}$ instead of the inner product between two vectors in $\mathbb{R}^n$. Formally, for every $j \in [s_\phi]$, it holds that

$$
\beta_j = \hat{\mathbf{A}}_{j*} \begin{bmatrix} \sum_{i=1}^{s_\phi} \alpha_i k\left(\mathbf{S}_{i*}, \mathbf{X}_{1*}\right) \\ \sum_{i=1}^{s_\phi} \alpha_i k\left(\mathbf{S}_{i*}, \mathbf{X}_{2*}\right) \\ \vdots \\ \sum_{i=1}^{s_\phi} \alpha_i k\left(\mathbf{S}_{i*}, \mathbf{X}_{n*}\right) \end{bmatrix} = \left[ \sum_{t=1}^{n} \hat{\mathbf{A}}_{j,t} k\left(\mathbf{S}_{1*}, \mathbf{X}_{t*}\right), \cdots, \sum_{t=1}^{n} \hat{\mathbf{A}}_{j,t} k\left(\mathbf{S}_{(s_\phi)*}, \mathbf{X}_{t*}\right) \right] \begin{bmatrix} \alpha_1 \\ \vdots \\ \alpha_{s_\phi} \end{bmatrix}.
$$

Thus, for every $j \in [s_\phi]$, define $\mathbf{A}_{j*} = \left[ \sum_{t=1}^{n} \hat{\mathbf{A}}_{j,t} k\left(\mathbf{S}_{1*}, \mathbf{X}_{t*}\right), \cdots, \sum_{t=1}^{n} \hat{\mathbf{A}}_{j,t} k\left(\mathbf{S}_{s_\phi*}, \mathbf{X}_{t*}\right) \right] \in \mathbb{R}^{s_\phi}$. In other words, $A$ is a the result of a Hadamard multiplication of $\hat{A}$ and a 1-rank matrix $G$ such that each of its rows is equal to $\left[ \sum_{t=1}^{n} k\left(\mathbf{S}_{1*}, \mathbf{X}_{t*}\right), \cdots, \sum_{t=1}^{n} k\left(\mathbf{S}_{s_\phi*}, \mathbf{X}_{t*}\right) \right]$.

Since the rank of $\hat{A}$ is full, i.e., $\mathrm{rank}\,(A) = \mathrm{rank}\left( \widetilde{\mathbf{X}}^T \left( \frac{1}{s_\phi} \widetilde{\mathbf{X}} \widetilde{\mathbf{X}}^T + n\lambda \mathbf{I}_n \right)^{-1} \right) = \mathrm{rank}\left( \widetilde{\mathbf{X}} \right) = s_\phi$ by assumption, then it holds that

$$
\mathrm{rank}\,(A) = \mathrm{rank}\left( \hat{A} \circ G \right) = \mathrm{rank}\left( D_v \hat{A} D_u \right) = \mathrm{rank}\left( \hat{A} \right),
$$

where the first equality holds by definition of $A$ and $\circ$ denoting the Hadamard multiplication product, the second inequality holds since by construction of $\mathbf{S}$ and $G = uv^T$ such that $u, v \in \mathbb{R}^{s_\phi}$ are vector with non-zero entries, and $D_u, D_v \in \mathbb{R}^{s_\phi \times s_\phi}$ are diagonal matrices where their diagonal are $u$ and $v$ respectively. The last inequality holds by property of rank function, i.e., the rank of any product of pair of square matrices $C$ and $D$ such that $D$ is of full rank is equal to the rank of $C$.

The right-hand side of (5) can reformulated as

$$
\frac{1}{s_\phi} \widetilde{\mathbf{X}}^T \left( \frac{1}{s_\phi} \widetilde{\mathbf{X}} \widetilde{\mathbf{X}}^T + n\lambda \mathbf{I}_n \right)^{-1} \mathbf{f}_{\mathbf{S}}[\mathbf{X}] = \mathbf{A}\alpha, \tag{6}
$$

where now we only need to solve $\beta = \mathbf{A}\alpha$. Such a linear system of equations has a solution since we have $s_\phi$ variables (the length of $\alpha$) and $s_\phi$ equations and the rank $A$ is equal to $s_\phi$. For simplicity, a solution to the above equality would be $\alpha := (\mathbf{A})^\dagger \beta$. To proceed in proving (ii) with all of the above ingredients, we utilize the following tool.

**Lemma 5** (Special case of Definition 6.1 from [BFL$^+$16]). *Let $X$ be a set, and let $\left( X, \|\cdot\|_2^2 \right)$ be a 2-metric space i.e., for every $x, y, z \in X$, $\|x - y\|_2^2 \le 2\left( \|x - z\|_2^2 + \|y - z\|_2^2 \right)$. Then, for every $\varepsilon \in (0, 1)$, and $x, y, z \in X$,*

$$
(1 - \varepsilon) \|y - z\|_2^2 - \frac{4}{\varepsilon^2} \|x - z\|_2^2 \le \|x - y\|_2^2 \le \frac{4}{\varepsilon^2} \|x - z\|_2^2 + (1 + \varepsilon) \|y - z\|_2^2. \tag{7}
$$

We note that Lemma 5 implies that $x, y, z \in \mathbb{R}^d$

$$
\|x - y\|_2^2 \le \min_{\tau \in \Upsilon} \max\left\{ \tau, \frac{4}{\tau^2} \right\} \|x - z\|_2^2 + \min\left\{ 1 + \tau, \frac{4(1 + \tau)}{3\tau} \right\} \|y - z\|_2^2. \tag{8}
$$

where for $\tau = 2$ we get the inequality associated with the property of 2-metric, and for any $\tau \in (0, 1)$, we obtain the inequality (7).

We thus observe that

$$\frac{1}{n}\sum_{i=1}^{n}\left|f_{[\mathbf{X},y,k]}^{\lambda}\left(\mathbf{X}_{i*}\right)-f_{[\mathbf{S},y_{\mathbf{S}},k]}^{\lambda,\mathbf{X},y}\left(\mathbf{X}_{i*}\right)\right|^{2}$$

$$=\frac{1}{n}\sum_{i=1}^{n}\left|f_{[\mathbf{X},y,k]}^{\lambda}\left(\mathbf{X}_{i*}\right)-f_{[\widetilde{\mathbf{X}},y,\phi]}^{\lambda}\left(\widetilde{\mathbf{X}}_{i*}\right)+f_{[\widetilde{\mathbf{X}},y,\phi]}^{\lambda}\left(\widetilde{\mathbf{X}}_{i*}\right)-f_{[\mathbf{S},y_{\mathbf{S}},k]}^{\lambda,\mathbf{X},y}\left(\mathbf{X}_{i*}\right)\right|^{2}$$

$$\leq\min_{\tau\in\Upsilon}\frac{\max\left\{\tau,\frac{4}{\tau^{2}}\right\}}{n}\sum_{i=1}^{n}\left|f_{[\mathbf{X},y,k]}^{\lambda}\left(\mathbf{X}_{i*}\right)-f_{[\widetilde{\mathbf{X}},y,\phi]}^{\lambda}\left(\widetilde{\mathbf{X}}_{i*}\right)\right|^{2}+$$

$$\frac{\min\left\{1+\tau,\frac{4(1+\tau)}{3\tau}\right\}}{n}\sum_{i=1}^{n}\left|f_{[\widetilde{\mathbf{X}},y,\phi]}^{\lambda}\left(\widetilde{\mathbf{X}}_{i*}\right)-f_{[\mathbf{S},y_{\mathbf{S}},k]}^{\lambda,\mathbf{X},y}\left(\mathbf{X}_{i*}\right)\right|^{2}$$

$$\leq\min_{\tau\in\Upsilon}2\max\left\{\tau,\frac{4}{\tau^{2}}\right\}\lambda+2\min\left\{1+\tau,\frac{4\left(1+\tau\right)}{3\tau}\right\}\lambda$$

$$=\min_{\tau\in\Upsilon}\left(2\max\left\{\tau,\frac{4}{\tau^{2}}\right\}+2\min\left\{1+\tau,\frac{4\left(1+\tau\right)}{3\tau}\right\}\right)\lambda,$$

where the first equality holds by adding and subtracting the same term, the first inequality holds by Lemma 5, and the second inequality holds by combining the way $f_{[\mathbf{S},y_{\mathbf{S}},k]}^{\lambda,\mathbf{X},y}$ was defined and Theorem 2. Finally, to conclude the proof of Theorem 3, we derive 4

$$\frac{1}{n}\sum_{i=1}^{n}\left|y_{i}-f_{[\mathbf{S},y_{\mathbf{S}},k]}^{\lambda,\mathbf{X},y}\left(\mathbf{X}_{i*}\right)\right|^{2}=\frac{1}{n}\sum_{i=1}^{n}\left|y_{i}-f_{[\widetilde{\mathbf{X}},y,\phi]}^{\lambda}\left(\widetilde{\mathbf{X}}_{i*}\right)+f_{[\widetilde{\mathbf{X}},y,\phi]}^{\lambda}\left(\widetilde{\mathbf{X}}_{i*}\right)-f_{[\mathbf{S},y_{\mathbf{S}},k]}^{\lambda,\mathbf{X},y}\left(\mathbf{X}_{i*}\right)\right|^{2}$$

$$\leq\min_{\tau\in\Upsilon}\frac{\min\left\{1+\tau,\frac{4(1+\tau)}{3\tau}\right\}}{n}\sum_{i=1}^{n}\left|y_{i}-f_{[\widetilde{\mathbf{X}},y,\phi]}^{\lambda}\left(\widetilde{\mathbf{X}}_{i*}\right)\right|^{2}$$

$$+\frac{\max\left\{\tau,\frac{4}{\tau^{2}}\right\}}{n}\sum_{i=1}^{n}\left|f_{[\widetilde{\mathbf{X}},y,\phi]}^{\lambda}\left(\widetilde{\mathbf{X}}_{i*}\right)-f_{[\mathbf{S},y_{\mathbf{S}},k]}^{\lambda,\mathbf{X},y}\left(\mathbf{X}_{i*}\right)\right|^{2}$$

$$\leq\min_{\tau\in\Upsilon}\frac{\min\left\{1+\tau,\frac{4(1+\tau)}{3\tau}\right\}}{n}\sum_{i=1}^{n}\left|y_{i}-f_{[\widetilde{\mathbf{X}},y,\phi]}^{\lambda}\left(\widetilde{\mathbf{X}}_{i*}\right)\right|^{2}+2\max\left\{\tau,\frac{4}{\tau^{2}}\right\}\lambda$$

$$\leq\min_{\tau\in\Upsilon}\frac{\min\left\{1+\tau,\frac{4(1+\tau)}{3\tau}\right\}}{n}\sum_{i=1}^{n}\left|y_{i}-f_{[\mathbf{X},y,k]}^{\lambda}\left(\mathbf{X}_{i*}\right)\right|^{2}$$

$$+\left(4\min\left\{1+\tau,\frac{4\left(1+\tau\right)}{3\tau}\right\}+2\max\left\{\tau,\frac{4}{\tau^{2}}\right\}\right)\lambda,$$

(9)

where the equality holds by adding and subtracting the same term, the first inequality holds by (8), and the second inequality follows as a result of the way $f_{\mathbf{S}}^{\lambda}$ was constructed and the fact that $\beta$ is its infimum based on Lemma 4, and the last inequality holds by Theorem 2. □

**Intuition behind Theorem 3.** The goal of Theorem 3 is to prove the existence of a small distilled set $\mathbf{S}$ (its size is a function of the minimal dimension of the RFF mapping required to ensure the provable additive approximation stated in Theorem 2) satisfying that: (i) The Ridge regression model trained on the mapped training data via RFF is identical to that of the Ridge regression model trained on the mapped small distilled set via RFF, (ii) more importantly there exists a KRR solution formulated for $\mathbf{S}$ with respect to the loss of the whole big data $\mathbf{X}$, which approximates the KRR solution on the whole data $\mathbf{X}$ (which is the goal of KRR-based dataset distillation techniques). Thus, (iii) we derive bounds on the difference (approximation error) between (1) The MSE between the ground truth labels of the full data and their corresponding predictions obtained by the specific KRR model (we previously described) on our distilled set and (2) The MSE between the ground truth labels and the predictions obtained when applying KRR on the whole data $\mathbf{X}$.

**The heart of our approach** lies in connecting the minimal dimension of the RFF required for provable additive approximation and the size of the distilled set. This is first done by showing that the distilled set can be any set $S$ of instances from the input space (e.g., images) and their corresponding labels, as long as the corresponding labels must maintain a certain property. Specifically speaking, the labels of the distilled set need to be in correlation with the normal of the best hyperplane found to fit the mapped training data via RFF $\widetilde{\mathbf{X}}$ via the Ridge regression model trained on $\left(\widetilde{\mathbf{X}}, y\right)$, i.e., $\left(\widetilde{\mathbf{S}}^T\widetilde{\mathbf{S}} + \lambda n s_\phi \lambda I_{s_\phi}\right)\left(\widetilde{\mathbf{X}}^T\widetilde{\mathbf{X}} + \lambda n s_\phi \lambda I_{s_\phi}\right)^{-1}\widetilde{\mathbf{X}}^T y = \widetilde{\mathbf{S}}^T y_{\mathbf{S}}$. From here, the idea hinges upon showing the existence of a KRR model (represented by a prediction function) that would be dependent on the prediction function that can be obtained from applying the Ridge regression problem to the mapped full training data via RFF. With such a model, the idea is to retrieve the predictions obtained when using the Ridge regression problem from the mapped training data via RFF via the use of some KRR model used on the distilled set. We thus show that through careful mathematical derivations, equation reformulation (involving ), and solving a system of equations, one is able to show the existence of a KRR solution that would allow us to use Theorem 2. Finally, to obtain our bounds, we also rely on the use of the weak triangle inequality. To that end, we now utilize the described KRR model on the distilled data together with Theorem 2 to achieve (iii).

**Remark 6.** *Note that the assumption that $\widetilde{\mathbf{X}}$ is of full rank (i.e., $s_\phi$) can be dropped easily from Theorem 3, and as a result, we obtain that $S$ can contain $r$ (rank of $\widetilde{\mathbf{X}}$) instances (rows of $\mathbf{S}$). For additional details, please refer to Section E in the Appendix.*

To simplify the bounds stated at Theorem 3, we provide the following remark.

**Remark 7.** *By fixing $\tau := 2$, the bounds in Theorem 3 become*

$$\frac{1}{n}\sum_{i=1}^{n}\left|f_{[\mathbf{X},y,k]}^{\lambda}\left(\mathbf{X}_{i*}\right) - f_{[\mathbf{S},y_{\mathbf{S}},k]}^{\lambda,\mathbf{X},y}\left(\mathbf{X}_{i*}\right)\right|^2 \leq 8\lambda,$$

*and*

$$\frac{1}{n}\sum_{i=1}^{n}\left|y_i - f_{[\mathbf{S},y_{\mathbf{S}},k]}^{\lambda,\mathbf{X},y}\left(\mathbf{X}_{i*}\right)\right|^2 \leq \frac{2}{n}\sum_{i=1}^{n}\left|y_i - f_{[\mathbf{X},y,k]}^{\lambda}\left(\mathbf{X}_{i*}\right)\right|^2 + 12\lambda.$$

*As for fixing $\tau := \varepsilon \in (0,1)$, we obtain that*

$$\frac{1}{n}\sum_{i=1}^{n}\left|y_i - f_{[\mathbf{S},y_{\mathbf{S}},k]}^{\lambda,\mathbf{X},y}\left(\mathbf{X}_{i*}\right)\right|^2 \leq \frac{1+\varepsilon}{n}\sum_{i=1}^{n}\left|y_i - f_{[\mathbf{X},y,k]}^{\lambda}\left(\mathbf{X}_{i*}\right)\right|^2 + \left(4\left(1+\varepsilon\right) + \frac{8}{\varepsilon^2}\right)\lambda.$$

## 5  Experimental Study

To validate our theoretical bounds, we performed distillation on three datasets: two synthetic datasets consisting of data generated from a Gaussian Random Field, and classification of two clusters, and one real dataset of MNIST binary and multi-class classification. Full experimental details for all experiments are available in the appendix.

**2d Gaussian Random Fields.** We first test our bounds by distilling data generated from the Gaussian Process prior induced by a kernel, $k$ on 2d data. We use a squared exponential kernel with lengthscale parameter $l = 1.5$: $k(x, x') = e^{-\frac{||x-x'||_2^2}{2l^2}}$. For $\mathbf{X}$, we sample $n = 10^5$ datapoints from $\mathcal{N}(0, \sigma_x^2)$, with $\sigma_x \in [0.25, 5.0]$. We then sample $y \sim \mathcal{N}(0, K_{XX} + \sigma_y^2 I_n)$, $\sigma_y = 0.01$. We fix $\lambda = 10^{-5}$ and distill down to $s = d_k^\lambda \log d_k^\lambda$. The resulting values of $d_k^\lambda$, $s$, and compression ratios are plotted in fig. 2. We additionally plot the predicted upper bound given by Remark 7 and the actual distillation loss. Our predicted upper bound accurately bounds the actual distillation loss. To better visualize how distillation affects the resulting KRR prediction, we show the KRR predictive function $f_{\mathbf{X}}^\lambda$ and the distilled predictive $f_{\mathbf{S}}^\lambda$ for $\sigma_x = 5.0$ in fig. 1b and fig. 1c.

**Two Gaussian Clusters Classification.** Our second synthetic dataset is one consisting of two Gaussian clusters centered at $(-2, 0)$ and $(2, 0)$, with labels $-1$ and $+1$, respectively. Each cluster contains 5000 datapoints so that $n = 10^5$. Each cluster as standard deviation $\sigma_x \in [0.25, 5.0]$. Additionally, two allow the dataset to be easily classified, we clip the $x$ coordinates of clusters 1 and

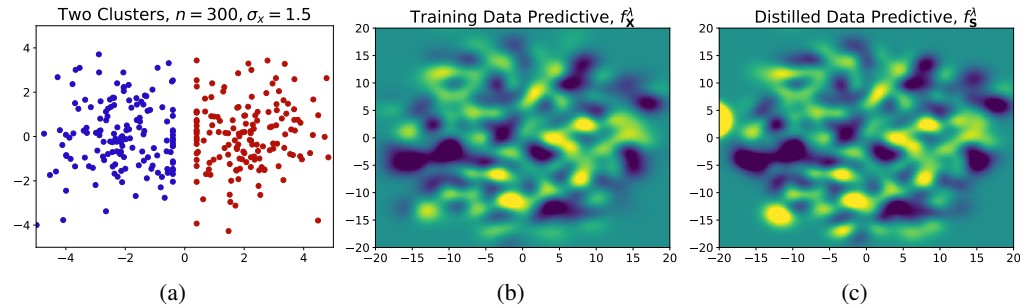

(a)             (b)             (c)

Figure 1: (a) visualizes the two clusters dataset with $n = 300$ and $\sigma_x = 1.5$. (b) and (c) visualizing the KRR predictive functions generated by the original dataset (b) and the distilled dataset (c) for the Gaussian Random Field experiment for $\sigma_x = 5.0$. The distilled dataset is able to capture all the nuances of the original dataset with a fraction of the datapoints.

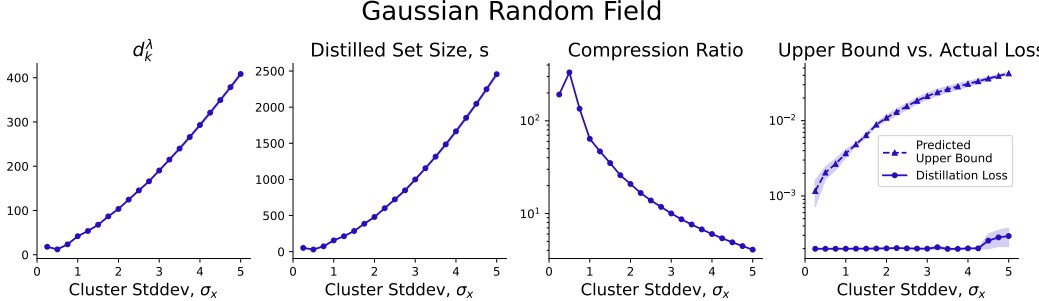

Figure 2: Distillation results for synthetic data generated by a Gaussian Random Field (n = 3)

clusters 2 to not exceed/drop below $-0.4$ and $0.4$, for the two clusters, respectively. This results in a margin between the two classes. We visualize the dataset for $n = 300$ and $\sigma = 1.5$ in fig. 1a. We use the same squared exponential kernel as before with $l = 1.5$, fix $\lambda = 10^{-5}$, and distill with the same protocol as before. We likewise plot $d_k^\lambda$, $s$, and compression ratios and distillation losses in fig. 3, again with our bound accurately containing the true distillation loss.

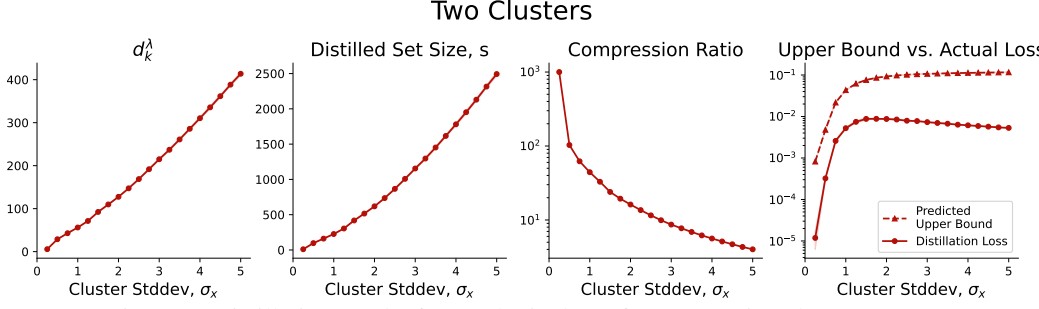

Figure 3: Distillation results for synthetic data of two Gaussian clusters (n = 3)

**Real Word Datasets Classification.** For our next test, we first consider binary classification on (i) MNIST 0 and 1 digits, (ii) SVHN 0 and 1 digit, and (iii) CIFAR-10 ship vs deer; all with labels $-1$ and $+1$, respectively. We use the same squared-exponential kernel with $l = 13.9$, $l = 3.0$, and $l = 8.0$, for MNIST, SVHN and CIFAR-10, respectively, which was chosen to maximize the marginal-log-likelihood, treating the problem as Gaussian Process regression. We vary $n \in [500, 10000]$, with an equal class split, and perform the same distillation protocol. Here, we additionally scale $\lambda \propto \frac{1}{\sqrt{n}}$ such that $\lambda = 10^{-4}$ when $n = 5000$. Distilling yields fig. 4, fig. 5, and fig. 6, showing that our bounds can accurately predict distillation losses for real-world datasets. In fig. 7 and fig. 8 in the appendix, we test our bound for the multi-class case on MNIST 0,1,2 digits with similar settings as in the binary classification case. The results justify our bounds.

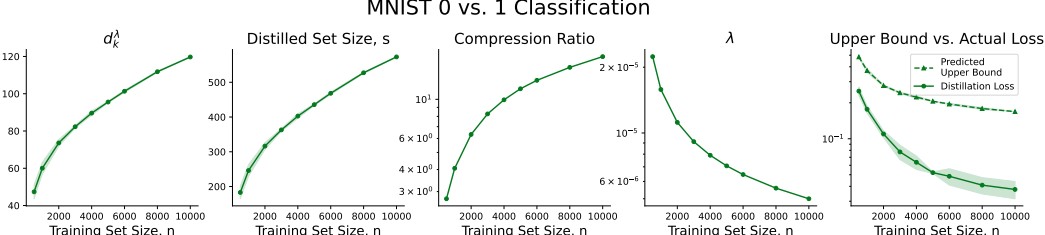

Figure 4: Distillation results for MNIST binary 0 vs. 1 classification (n = 3)

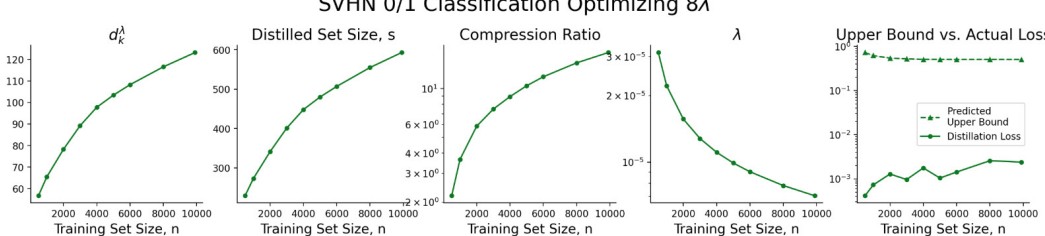

Figure 5: Distillation results for SVHN binary 0 vs. 1 classification (n = 3)

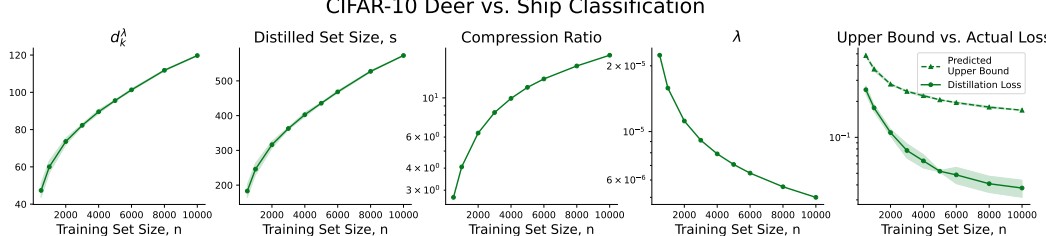

Figure 6: Distillation results for CIFAR-10 binary deer vs. ship classification (n = 3)

## 6   Conclusion

In this study, we adopt a theoretical perspective to provide bounds on the (sufficient) size and approximation error of distilled datasets. By leveraging the concept of random Fourier features (RFF), we prove the existence of small distilled datasets and we bound their corresponding excess risk when using shift-invariant kernels. Our findings indicate that the size of the guaranteed distilled data is a function of the "number of effective degrees of freedom," which relies on factors like the kernel, the number of points, and the chosen regularization parameter, $\lambda$, which also controls the excess risk. In particular, we demonstrate the existence of a small subset of instances within the original input space, where the solution in the RFF space coincides with the solution found using the input data in the RFF space. Subsequently, we show that this distilled subset of instances can be utilized to generate a KRR solution that approximates the KRR solution obtained from the complete input data. To validate these findings, we conducted empirical examinations on both synthetic and real-world datasets supporting our claim. While this study provides a vital first step in understanding the theoretical limitations of dataset distillation, the proposed bounds are not tight, as seen by the gap between the theoretical upper bound and the empirical distillation loss in section 5. Future work could look at closing this gap, as well as better understanding the tradeoff between distillation size and relative error.

## acknowledgements

This research has been funded in part by the Office of Naval Research Grant Number Grant N00014-18-1-2830, DSTA Singapore, and the J. P. Morgan AI Research program.

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

# A Experiment Details

All experiments unless otherwise stated present the average/standard deviations of $n = 3$ runs. Each run consists of a random subset of MNIST 0/1 digits for MNIST binary classification, or random positions of sampled datapoints for synthetic data, and different samples from the GP for the Gaussian Random Field experiment. Distilled datasets are initialized as subsets of the original training data. We distill for 20000 iterations with Adam optimizer with a learning rate of 0.002 optimizing both images/data positions and labels. We use full batch gradient descent for the synthetic datasets and a maximum batch size of 2000 for the MNIST experiment. For the MNIST experiment we found that particularly for larger values of $n$, with minibatch training, we could obtain lower distillation losses by optimizing for longer, so the closing of the gap between the upper bound and experiment values in fig. 4 may be misleading: longer optimization could bring the actual distillation loss lower.

To ensure that assumption (II) is fulfilled, we scale the labels such that $\left\| f^\lambda_{[\mathbf{X},y,k]} \right\|_{\mathcal{H}} = 1$. For example, if we are working with MNIST binary classification, with labels $\{+1, -1\}$, we first compute $\left\| f^\lambda_{[\mathbf{X},y,k]} \right\|_{\mathcal{H}} = r$ using $\{+1, -1\}$ labels, then rescale the labels by $1/r$ so that the labels are $\{+\frac{1}{r}, -\frac{1}{r}\}$. Suppose this results in some upper bound $\mathcal{L}_U$ and some real distillation loss $\mathcal{L}_R$. For the corresponding plots in figs. 2 to 4, we plot $r^2 \mathcal{L}_U$ and $r^2 \mathcal{L}_R$. We do this because the $r$ values for different parameters (such as $n$ or $\sigma_x$) could be different, and scaling for the plots allows the values to be comparable.

In the figures for the upper bounds on the distillation loss we plot the smallest value of the upper bounds in remark 7.

## A.1 Code

Code is available in the supplementary material.

# B Proof of Technical Results

## B.1 Proof of Theorem 2

Note that the proof of Theorem 2 follows from the proofs of Theorem 1 and Corollary 1 of [LTOS21]. For completeness, we provide the following proof.

**Theorem 2** (A result of the proof of Theorem 1 and Corollary 2 of [LTOS21]). *Let* $\mathbf{X} \in \mathbb{R}^{n \times d}$ *be an input matrix,* $y \in \mathbb{R}^n$ *be an input label vector,* $k : \mathbb{R}^d \times \mathbb{R}^d \to [0, \infty)$ *be a shift-invariant kernel function, and* $\mathbf{K} \in \mathbb{R}^{n \times n}$, *where* $\forall i, j \in [n] : \mathbf{K}_{i,j} = k(\mathbf{X}_{i*}, \mathbf{X}_{j*})$. *Let* $\lambda > 0$, *and let* $d^\lambda_{\mathbf{K}} = Tr\left( \mathbf{K}\left(\mathbf{K} + n\lambda\mathbf{I}_n\right)^{-1}\right)$. *Let* $s_\phi \in \Omega\left(d^\lambda_{\mathbf{K}} \log\left(d^\lambda_{\mathbf{K}}\right)\right)$ *be a positive integer. Then, there exists a pair* $(\phi, \widetilde{\mathbf{X}})$ *such that (i)* $\phi$ *is a mapping* $\phi : \mathbb{R}^d \to \mathbb{R}^{s_\phi}$ *(which is based on either the weighted RFF function or the RFF function [LTOS21]), (ii)* $\widetilde{\mathbf{X}}$ *is a matrix* $\widetilde{\mathbf{X}} \in \mathbb{R}^{n \times s_\phi}$ *where for every* $i \in [n]$, $\widetilde{\mathbf{X}}_{i*} := \phi\left(\mathbf{X}_{i*}\right)$, *and (iii)* $(\phi, \widetilde{\mathbf{X}})$ *satisfies*

$$\frac{1}{n}\sum_{i=1}^{n}\left| y_i - f^\lambda_{\left[\widetilde{\mathbf{X}}, y, \phi\right]}\left(\widetilde{\mathbf{X}}_{i*}\right)\right|^2 \leq \frac{1}{n}\sum_{i=1}^{n}\left| y_i - f^\lambda_{[\mathbf{X}, y, k]}\left(\mathbf{X}_{i*}\right)\right|^2 + 4\lambda,$$

*where* $f^\lambda_{\left[\widetilde{\mathbf{X}}, y, \phi\right]} : \mathbb{R}^{s_\phi} \to \mathbb{R}$ *such that for every row vector* $z \in \mathbb{R}^{s_\phi}$, $f^\lambda_{\left[\widetilde{\mathbf{X}}, y, \phi\right]}(z) = z\left(\widetilde{\mathbf{X}}^T\widetilde{\mathbf{X}} + \lambda n s_\phi \lambda \mathbf{I}_{s_\phi}\right)^{-1}\widetilde{\mathbf{X}}^T y$. *Note that, Table 1 gives bounds on* $s_\phi$ *when* $\lambda \propto \frac{1}{\sqrt{n}}$.

*Proof.* First, let $\beta \in \mathbb{R}^{s_\phi}$ be the ridge regression solution involving $\widetilde{\mathbf{X}}$ and $y$, i.e.,

$$\beta \in \underset{x \in \mathbb{R}^{s_\phi}}{\arg\min}\ \frac{1}{n}\sum_{i=1}^{n}\left| y_i - \widetilde{\mathbf{X}}_{i*}x\right|^2 + \lambda s\left\|x\right\|_2^2.$$

Note that, by construction, for every $i \in [n]$, $f^\lambda_{\left[\widetilde{\mathbf{X}}, y, \phi\right]}\left(\widetilde{\mathbf{X}}_{i*}\right)$ is equal to $\widetilde{\mathbf{X}}_{i*} \cdot \beta$.

Thus,

$$\frac{1}{n}\sum_{i=1}^{n}\left|y_i - f^{\lambda}_{[\widetilde{\mathbf{X}},y,\phi]}\left(\widetilde{\mathbf{X}}_{i*}\right)\right|^2 = \frac{1}{n}\sum_{i=1}^{n}\left|y_i - \widetilde{\mathbf{X}}_{i*}\beta\right|^2 = \inf_{\|\widetilde{f}\|_{\mathcal{H}}\leq\sqrt{2}}\frac{1}{n}\sum_{i=1}^{n}\left|y_i - \widetilde{f}\left(\mathbf{X}_{i*}\right)\right|^2 \quad (10)$$

where the first equality holds by definition of $f^{\lambda}_{[\widetilde{\mathbf{X}},y,\phi]}\left(\widetilde{\mathbf{X}}_{i*}\right)$ for every $i \in [n]$, and the second equality follows from the proof of Lemma 6 of [LTOS21] which indicates that $s\|\beta\|_2^2 \leq 2$.

Note that,

$$\inf_{\|\widetilde{f}\|_{\mathcal{H}}\leq\sqrt{2}}\frac{1}{n}\sum_{i=1}^{n}\left|y_i - \widetilde{f}\left(\mathbf{X}_{i*}\right)\right|^2$$

$$= \inf_{\|\widetilde{f}\|_{\mathcal{H}}\leq\sqrt{2}}\frac{1}{n}\sum_{i=1}^{n}\left|y_i - f^{\lambda}_{[\mathbf{X},y,k]}\left(\mathbf{X}_{i*}\right) + f^{\lambda}_{[\mathbf{X},y,k]}\left(\mathbf{X}_{i*}\right) - \widetilde{f}\left(\mathbf{X}_{i*}\right)\right|^2$$

$$= \inf_{\|\widetilde{f}\|_{\mathcal{H}}\leq\sqrt{2}}\frac{1}{n}\sum_{i=1}^{n}\left|y_i - f^{\lambda}_{[\mathbf{X},y,k]}\left(\mathbf{X}_{i*}\right)\right|^2 + \frac{1}{n}\sum_{i=1}^{n}\left|\widetilde{f}\left(\mathbf{X}_{i*}\right) - f^{\lambda}_{[\mathbf{X},y,k]}\left(\mathbf{X}_{i*}\right)\right|^2 \quad (11)$$

$$+ \frac{2}{n}\sum_{i=1}^{n}\left(y_i - f^{\lambda}_{[\mathbf{X},y,k]}\left(\mathbf{X}_{i*}\right)\right)\left(f^{\lambda}_{[\mathbf{X},y,k]}\left(\mathbf{X}_{i*}\right) - \widetilde{f}\left(\mathbf{X}_{i*}\right)\right)$$

where the first equality holds by adding and subtracting the same amount, while the last equality holds by simple expansion.

To properly bound the terms above, we rely on Lemma 7 of [LTOS21].

**Lemma 8** (Restatement of Lemma 7 [LTOS21]). *Under Assumption 1 and the definitions in Theorem 2, and for*

$$\widehat{f}^{\lambda} \in \arg\min_{\widetilde{f}\in\widetilde{\mathcal{H}}}\frac{1}{n}\sum_{i=1}^{n}\left|f^{\lambda}_{[\mathbf{X},y,k]}\left(\mathbf{X}_{i*}\right) - \widetilde{f}\left(\mathbf{X}_{i*}\right)\right|^2 + \lambda\left\|\widetilde{f}\right\|_{\widetilde{\mathcal{H}}},$$

*where $\widetilde{\mathcal{H}}$ is the corresponding RKHS space of $\widetilde{\mathbf{X}}$, it holds that*

$$\frac{1}{n}\sum_{i=1}^{n}\left(y_i - f^{\lambda}_{[\mathbf{X},y,k]}\left(\mathbf{X}_{i*}\right)\right)\left(f^{\lambda}_{[\mathbf{X},y,k]}\left(\mathbf{X}_{i*}\right) - \widehat{f}^{\lambda}\left(\mathbf{X}_{i*}\right)\right) \leq \lambda.$$

Hence, combining Lemma 8 with (11) yields

$$\inf_{\|\widetilde{f}\|_{\mathcal{H}}\leq\sqrt{2}}\frac{1}{n}\sum_{i=1}^{n}\left|y_i - \widetilde{f}\left(\mathbf{X}_{i*}\right)\right|^2$$

$$\leq \frac{1}{n}\sum_{i=1}^{n}\left|y_i - f^{\lambda}_{[\mathbf{X},y,k]}\left(\mathbf{X}_{i*}\right)\right|^2 + 2\lambda + \inf_{\|\widetilde{f}\|_{\mathcal{H}}\leq\sqrt{2}}\frac{1}{n}\sum_{i=1}^{n}\left|\widetilde{f}\left(\mathbf{X}_{i*}\right) - f^{\lambda}_{[\mathbf{X},y,k]}\left(\mathbf{X}_{i*}\right)\right|^2 \quad (12)$$

$$\leq \frac{1}{n}\sum_{i=1}^{n}\left|y_i - f^{\lambda}_{[\mathbf{X},y,k]}\left(\mathbf{X}_{i*}\right)\right|^2 + 4\lambda$$

where the last inequality holds by Lemma 4 (or equivalently Lemma 6 of [LTOS21]), and thus concluding Theorem 2.

$\square$

## C   Regarding $s_\phi$

In what follows, we discuss cases for determining the quantity $s_\phi$. To quantify $s_\phi$ which in turn determines the size of the distilled set, we need to measure $d_K$ which is the trace of $\mathbf{K}\left(\mathbf{K} + n\lambda I_n\right)^{-1}$:

Table 1: Table 1 from [LTOS19]. The trade-off in the worst case for the squared error loss.

| SAMPLING SCHEME | SPECTRUM | NUMBER OF FEATURES |
|---|---|---|
| WEIGHTED RFF | finite rank | $s_\phi \in \Omega(1)$ |
| | $\lambda_i \propto A^i$ | $s_\phi \in \Omega(\log n \cdot \log \log n)$ |
| | $\lambda_i \propto i^{-2t} \ (t \geq 1)$ | $s_\phi \in \Omega(n^{1/2t} \cdot \log n)$ |
| | $\lambda_i \propto i^{-1}$ | $s_\phi \in \Omega(\sqrt{n} \cdot \log n)$ |
| PLAIN RFF | finite rank | $s_\phi \in \Omega(\sqrt{n})$ |
| | $\lambda_i \propto A^i$ | $s_\phi \in \Omega(\sqrt{n} \cdot \log \log n)$ |
| | $\lambda_i \propto i^{-2t} \ (t \geq 1)$ | $s_\phi \in \Omega(\sqrt{n} \cdot \log n)$ |
| | $\lambda_i \propto i^{-1}$ | $s_\phi \in \Omega(\sqrt{n} \cdot \log n)$ |

- For the case where $K$ has a finite rank, i.e., the number of positive eigenvalues is lower than $n$, then $s_\phi \in \Omega(1)$.

- As for the exponential decay, it occurs when the kernel is Gaussian and the marginal distribution of the input data (e.g., images) is sub-Gaussian. In such a case, it was shown in [Bac17] that $d_K \in O\left(\log \frac{1}{\lambda}\right)$. Thus $s_\phi$ is poly-logarithmic in $n$ when $\lambda := O\left(\frac{1}{\sqrt{n}}\right)$.

- For the case where the Hilbert space $\mathcal{H}$ is also a Sobolov space of order $\gamma$ larger or equal to 1, then $d_K \in O\left(\frac{1}{\lambda^{2\gamma}}\right)$ which in the case of $\lambda := O\left(\frac{1}{\sqrt{n}}\right)$, we have $s_\phi \in \Omega\left(n^{\frac{1}{4\gamma}}\right)$.

- In the general case, where the decay of the eigenvalues admits $\lambda_i \propto O(i^{-1})$, then $d_k$ in the worst case is bounded by $O(\frac{1}{lambda})$, and thus, via Theorem 2, we can deduce that $s_\phi \in O(\sqrt{n} \log n)$ for the case of $\lambda \in O(\frac{1}{\sqrt{n}})$.

Note that the choice of $\lambda \in O(\frac{1}{\sqrt{n}})$ is used in the literature of KRR to ensure that the learning rate is $\frac{1}{\sqrt{n}}$ as shown in [RR17].

# D  Multi-class Experiments

In fig. 7 and fig. 8 we test our bound for the multi-class case on MNIST 0,1,2 digits with similar settings as in the binary classification case. The results justify our bounds.

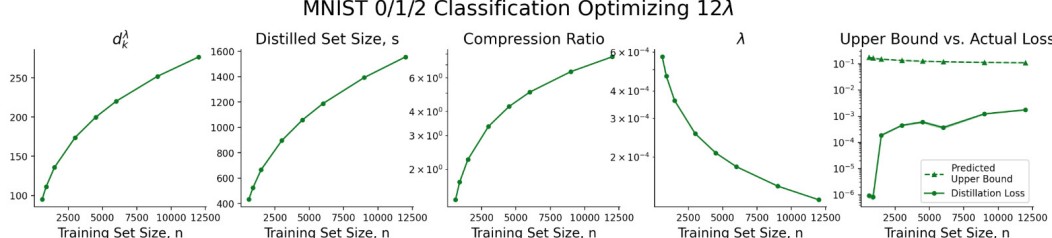

Figure 7: Distillation results for MNIST multi class 0, 1, 2 classification (n = 3)

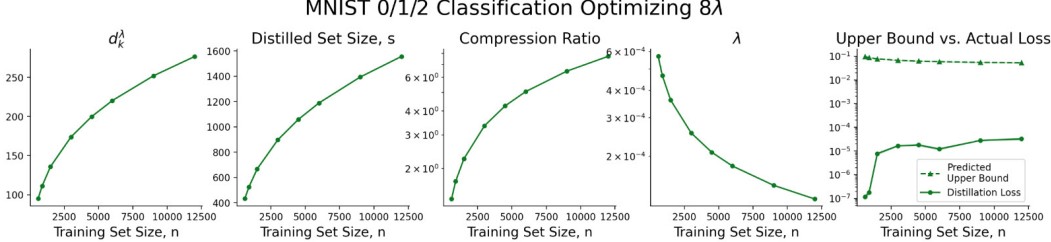

Figure 8: Distillation results for MNIST multi class 0, 1, 2 classification (n = 3); testing the $8\lambda$ bound

# E   On the rank of $\widetilde{\mathbf{X}}$ and its connection to the distilled set $S$

First, recall that $\widetilde{\mathbf{X}}$ is the RFF image of $X$. There are mainly two cases connecting the rank of $\widetilde{\mathbf{X}}$ to $S$:

(I) The rank of $\widetilde{\mathbf{X}}$ is $s_\phi$, i.e., $\widetilde{\mathbf{X}}$ is of full rank.

(II) Otherwise.

**Regarding Case (I).**   For such a case, Theorem 3 is exactly what we need to prove the existence of a distilled set $S$.

**Handling Case (II).**   For this case as well, one can also choose to ensure that the rank of $\mathbf{S}$ (matrix form of $S$) is $s_\phi$ (full rank) and apply the same derivations done in the proof of Theorem 3 without any violation from the perspective of the validity of our proof. However, due to the nature of this case, one can also ensure that the size of $S$ (number of instances) is much smaller than $s_\phi$, specifically, $r$ (the rank of $\widetilde{\mathbf{X}}$). To that end, must ensure that

(A)  the following equation

$$\left(\widetilde{\mathbf{S}}^T\widetilde{\mathbf{S}} + \lambda n s_\phi \lambda \mathbf{I}_{s_\phi}\right) \left(\widetilde{\mathbf{X}}^T\widetilde{\mathbf{X}} + \lambda n s_\phi \lambda \mathbf{I}_{s_\phi}\right)^{-1} \widetilde{\mathbf{X}}^T y = \widetilde{\mathbf{S}}^T y_{\mathbf{S}},$$

has a solution, i.e., $y_{\mathbf{S}}$ exists, and also

(B)  we need to ensure that (6) also has a solution, i.e., there exists $\alpha \in \mathbb{R}^r$ such that

$$\frac{1}{s_\phi}\widetilde{\mathbf{X}}^T \left(\frac{1}{s_\phi}\widetilde{\mathbf{X}}\widetilde{\mathbf{X}}^T + n\lambda \mathbf{I}_n\right)^{-1} \mathbf{f}_{\mathbf{S}}[\mathbf{X}] = \mathbf{A}\alpha.$$

We observe that since $\widetilde{\mathbf{X}}$ is rank deficient adds some obstacles to our proofs; when attempting to ensure both (A) and (B). Specifically speaking, the rank of $\left(\widetilde{\mathbf{S}}^T\widetilde{\mathbf{S}} + \lambda n s_\phi \lambda \mathbf{I}_{s_\phi}\right) \left(\widetilde{\mathbf{X}}^T\widetilde{\mathbf{X}} + \lambda n s_\phi \lambda \mathbf{I}_{s_\phi}\right)^{-1} \widetilde{\mathbf{X}}^T$ is equal to the rank of $\widetilde{\mathbf{X}}$ (due to properties of rank function) and since we aim to maintain that $S$ has exactly that number of rows while also its RFF image $\widetilde{\mathbf{S}}$ has a rank equal to that of $\widetilde{\mathbf{X}}$, we obtain that the linear system of equations in (A) might have no solution at all. This is unless the row span of $\widetilde{\mathbf{S}}$ coincides with that of $\widetilde{\mathbf{X}}$.

Having an $S$ such that the row span of its RFF image coincides with the row span of $\widetilde{\mathbf{X}}$, also ensures the existence of a solution concerning the linear system of equations present at (B).

# F   One of many ways to construct $S$

Following the previous section, to properly construct $S$ that would satisfy the proof of Theorem 3, we discuss two possible constructions:

- If $\tilde{X}$ is of full rank (the rank is $s_\phi$), then there exists a subset $S$ of the rows of $X$ such that its image in the RFF space is a linearly independent subset $\tilde{S}$ of the rows of $\tilde{X}$. Such a subset is of full rank as it is linearly independent. Such an algorithm can be accessed via [J23].

- If $\tilde{X}$ is rank deficient, i.e., $r < s_\phi$ where $r$ denotes the rank of $\tilde{X}$, then the problem by it self requires $r$ independent vectors, i.e, one can represent $\left(\tilde{X}^T\tilde{X} + \lambda n s_\phi \lambda I_{s_\phi}\right)^{-1} \tilde{X}^T y$ by $Az$ where $A \in \mathbb{R}^{s_\phi \times r}$ and $z \in \mathbb{R}^r$. This is since the rank of $\left(\tilde{X}^T\tilde{X} + \lambda n s_\phi \lambda I_{s_\phi}\right)^{-1} \tilde{X}^T$ is at max the minimum between the rank of $\left(\tilde{X}^T\tilde{X} + \lambda n s_\phi \lambda I_{s_\phi}\right)^{-1}$ and the rank of $\tilde{X}^T$. Such rank is bounded from above by $r$ since the rank of $\left(\tilde{X}^T\tilde{X} + \lambda n s_\phi \lambda I_{s_\phi}\right)^{-1}$ is $s_\phi$.

To this end, we can invoke the first case with respect to $(B, z)$. Thus, by choosing $S$ to contain the $r$ linearly independent rows in $\tilde{X}$ and an additional row from $\tilde{X}$ (not already in $S$), there exists $y_S$ such that,

$$\left( \tilde{S}^T \tilde{S} + \lambda n s_\phi \lambda I_{s_\phi} \right) Bz = \tilde{S}^T y_{\mathbf{S}}.$$

**Connection to subset selection and label solve.** Observe that according to this section, we showed that our bounds in Theorem 3 hold for a subset of the data but with different (potentially learned) labels (not necessarily from the set of input labels) which can be thought of as a hybrid between distillation and subset selection. Intuitively, this justifies the label-solve approach of [NCL20], where here we showed that a subset of the data can achieve similar bounds but with learning the labels.

