# OpenReview forum: "On the Size and Approximation Error of Distilled Datasets"
_NeurIPS.cc/2023/Conference — NeurIPS 2023 poster_

### Official Review · Reviewer_uCrm · 2023-06-25

**Soundness:** 3 good
**Presentation:** 3 good
**Contribution:** 2 fair
**Rating:** 5
**Confidence:** 4

**Summary:**

This paper provides theoretical analysis towards recent errors of recent dataset distillation methods based on kernel ridge regression (KRR).  It mainly utilizes some previous results on KRR and applies them in the context of dataset distillation. Some simple simulation results verify the derived bounds.

**Strengths:**

1. The paper derives valid error bounds for KRR-based dataset distillation methods under some assumptions. It provides a guidance for the size of synthetic dataset that is required to result in pleasant errors.
2. The writing is logical and consistent. The contributions of this paper and how this paper inherits previous results is very clear.

**Weaknesses:**

1. My major concern is that the analysis is not consistent with the typical setting of dataset distillation:
    * This paper uses a distilled dataset whose size is greater than the number of features encoded by the kernel. However, in practice, the KRR-based methods in dataset distillation often rely on infinite-wide neural networks and in practice this is achieved by a large feature dimension, while the size of distilled dataset is in fact very small, e.g., only 1 image per class.
    * In fact, the setting that the number of distilled samples is larger than the feature dimension makes the problem much simpler according my own experience. For example, we can directly solve the label $y$ without touching the samples $X$ as shown in Proof of (i) to get exactly the same KRR solution. Intuitively, if KRR solutions of real and synthetic datasets are consistent, predictions on data should also be consistent, which indicates that the results in the paper are somewhat trivial.
    * Moreover, this paper assumes that the output dimension is only 1. However, the output dimension is in fact the number of classes for typical classification problems. It is unclear whether the error bounds derived by the current paper still useful in real cases.
2. The technical contribution is not enough for a NeurIPS paper. The paper directly applies results of previous works. The results in the current paper can be viewed as an implication of previous results on the dataset distillation setting, since the setting is simple as mentioned previously.
3. The error bounds are not tight as shown in the experiment section, which means the theoretical results are not very useful in practice.
4. The experiments are too simple. The largest scale is only binary classification on MNIST dataset. For dataset distillation, at least results on CIFAR10 are expected as very standard benchmarks.

**Questions:**

1. The analysis for cases that the number of synthetic samples is smaller than the feature dimension.
2. The analysis for cases that the output dimension is larger than 1.
3. Experimental analysis on larger datasets.

Please refer to the weaknesses part for details.

**Limitations:**

I have not found issues related to this part.

---

> ### Author Rebuttal · Authors · 2023-08-09
>
> We greatly appreciate the insights shared by the reviewer and the expert evaluation they provided. Integrating their feedback, have already made significant improvements to the paper. We are looking forward to further engagement with the reviewer as we enter the upcoming open discussion phase.
>
> We have taken careful consideration of every comment raised in the review, ensuring a thorough response to each. We are optimistic that our comprehensive explanations may encourage the reviewer to consider raising the score. If any further clarification is required, please do not hesitate to reach out to us.
>
> *Comment 1.1:* The size of the distilled set is not related to the “feature space” dimension correlated with the NTK of infinite-wide networks or the dimension of the input. It is proportional to the dimension that is guaranteed by the “weighted RFF” of [1]. Thus, it is not necessarily large and depends mostly on the eigenvalues’ decay rate of the Gram matrix $K$ that corresponds to the kernel. For example (for shift-invariant kernels), if the decay rate is exponential, the size of the distilled set is polylogarithmic in $n$ or even $O(1)$ when $K$ has finite rank.
>
> *Comment 1.2:* We do not ensure that the KRR solution for $(\mathbf{S},y_{\mathbf{S}})$ is exactly the same KRR solution on the full data. In fact, the distilled set KRR solution is formulated via instances of the distilled set where the corresponding in-sample prediction function lies in a ball of radius $2\lambda$ from the in-sample prediction function of the solution $\beta$ to the Ridge regression (RR) model optimized for the RFF image of the full data, i,e., in our derivations, we aimed to find another in-sample prediction function in the KRR space involving $S$ such that it lies within a bounded space that would also encapsulate the optimal KRR found on the full data.
>
> Finally, the KRR objective function involves a weight vector $\alpha$, which is used to define the in-sample prediction of any $x\in X$ as a linear combination between each instance in the data ($S$). Thus the solution of the KRR on the entire data is not the same solution that we obtain due to the fact that these two solutions lie in two different spaces (different dimensionalities), and thus the in-sample prediction function is not guaranteed to be the same.
>
> *Comment 1.3:* The theory behind our approach can be used for the case of multiclass classification as kindly described in [1]. For multi-output data, $\beta$ will be a matrix where our theory will hold for every column in $\beta$ with the corresponding column of the labels matrix. We have expanded on this in our paper.
>
> *Comment 2:*  We thank the reviewer for the honesty. However, we respectfully disagree with those claims.  This result is indeed **not** an implication of previous results. While we did rely on tools from [1], the novelty of our paper lies in carefully analyzing the implied meaning of such tools and exploiting their underlying structure to our benefit.
>
> Specifically, we showed how to design the labels of the distilled set $S$ such that certain properties will hold, i.e., we showed that we can construct a label function for $S$ where we applied variable equation reformulation to enable solving a system of equations that would (a) yield that the label function maintains a direct connection to the solution of the RR problem involving the RFF image of the input data $X$, and (b) hinge upon the minimal size of $S$ to ensure the existence of such a label vector is dependent on the dimension of the RFF space that was investigated by [1].
>
> From there, we further ensure the existence of a KRR solution by showing that there exists a different in-sample prediction function that is close to the solution of the aforementioned solution to the RR problem involving the RFF space of $X$ in terms of the quality of the prediction. Such existence is indeed not trivial and requires careful analysis to ensure the existence of such a trait, e.g., (a) ensuring that the minimal distilled set needs to be larger than the dimension of the RFF space to enable the existence of a label vector corresponding to a distilled set that can be any set of instances (b) maneuvering through the properties that this label vector hinges upon exploiting the underlying structure associated with such vector to ensure the existence of an in-sample prediction function that resides in the same space that the optimal in-sample prediction function which is defined by the solution of the KRR model on  $X$. At first glance, this is not an immediate pass, and it did require careful derivation and ensuring certain properties hold.
>
> Our analysis did not stop here but rather we also bounded the added error of using our distilled set leading to a different bound to that of [1]. Note that in our context, no prior paper suggested any provable guarantee on the size of the distilled set or its approximation error; our paper is the first.
>
> *Comment 3:* Initially, our proofs are crucial for advancing the field in practice. By setting bounds on size and approximation, we offer a way to analyze, validate, and debug the implementation and correctness of new algorithms when used with unexplored datasets.
>
> Exploring the underlying theory supporting small distilled sets, substantiated by rigorous proofs, is a crucial milestone. This marks an initial step towards creating provably robust dataset distillation techniques, an aspect lacking in the existing literature. This is by using the foundational understanding of dataset distillation started in this work.  Hence, our work can be seen as the primary stride towards such a goal.  While more investigation is necessary for a comprehensive grasp of data distillation, this work serves as the first step in introducing theoretical guarantees to dataset distillation.
>
> *Comment 4:* following your valuable comment, we added experiments on Cifar10 (attached in PDF). Results on SVHN will be added.
>
> [1] = [LTOS21]

---

> > ### Comment · Reviewer_uCrm · 2023-08-11
> > **Thanks for the Rebuttal**
> >
> > I would like to thank the authors for the detailed response to the concerns and questions. Frankly speaking, I cannot easily follow every single line of theoretical derivation and the authors' rebuttal. What I discuss here is based on my understanding. If it is wrong please correct me.
> >
> > *Comment 1.1:* I have understood that the size of the distilled set is not related to the “feature space” dimension. However, I am very curious about how to **intuitively** understand the feature space dimension I mentioned in the comments, the eigenvalues’ decay rate of the Gram matrix mentioned by the authors, and their relationships. For example, in what cases the decay rate is exponential? Do the input datasets have to be equipped with some specific patterns?
> >
> > *Comment 1.2:* Looking at (1) of Theorem 3 again, I assume that $\lambda$ should be small and not make a large difference in the numerical results. If that is the case and we want to find the optimal label $y_S$, since the number of samples ($s_{\phi}+1$) is larger than the RFF features ($s_[\phi]$), the equation is underdetermined and we can definitely find proper $y_S$ to guarantee the same KRR solution. Even though it is not exactly the same due to the effect of $\lambda$ for regularization, the error should be small enough. If the error on KRR solutions is not large, we can expect that the error on predictions is not large, either. So I am not sure why the following large paragraph of theoretical analysis is necessary.
> >
> > *Comment 1.3:* I have understood that the theoretical analysis is also applicable to multi-class cases. However, I wonder if it is possible for the authors to validate this through experiments, to better see the scalability of the proposed theoretical tool. Please do not worry if the authors think the time is tight and I will definitely understand it. I just want to express my curiosity here.
> >
> > *Comment 2:* I have understood that the proposed theoretical analysis is not a trivial implication of previous results.
> >
> > *Comment 3:* I definitely understand that the paper is the first that focuses on theoretical analysis of the error bounds in dataset distillation. But indeed the gap is too large, especially for the case of small synthetic datasets. If the authors can provide some insightful explanation on this and discuss some possible future solutions, I think it is acceptable given that the paper is the first work on this direction.
> >
> > *Comment 4:* Thanks for the new results!
> >
> > Overall, I choose increase my score to 4 given that the authors addressed part of my concerns.

---

> > > ### Author Response · Authors · 2023-08-11
> > > **Thank you + additional answers**
> > >
> > > We thank the reviewer for the quick response and for engaging with us. Indeed we appreciate that you raised your score. It is great to see how the open review process is beneficial this way.
> > >
> > >
> > > **Comment 1.1:**
> > >
> > >  We thank the reviewer for pointing this out. Here are some deeper details – in order to quantify $s_\phi$ which in turn determines the size of the distilled set, we need to measure $d_K$ which is the trace of $\mathbf{K} \left( \mathbf{K} + n\lambda I_n\right)^{-1}$:
> > >  1. For the case where $K$ has a finite rank, i.e., the number of positive eigenvalues is lower than $n$, then $s_\phi \in \Omega(1)$.
> > >  2. As for the exponential decay, it occurs when the kernel is Gaussian and the marginal distribution of the input data (e.g., images) is sub-Gaussian. In such a case, it was shown in [1] that $d_K \in O\left(\log{\frac{1}{\lambda}} \right)$. Thus $s_\phi$ is poly-logarithmic in $n$ when $\lambda := O\left( \frac{1}{\sqrt{n}}\right)$.
> > >  3. For the case where the Hilbert space $\mathcal{H}$ is also a Sobolov space of order $\gamma$ larger or equal to 1, then $d_K \in O\left(\frac{1}{\lambda^{2\gamma}}\right)$ which in the case of $\lambda := O\left( \frac{1}{\sqrt{n}}\right)$, we have $s_\phi \in \Omega\left(n^{\frac{1}{4\gamma}}\right)$.
> > >  4. In the general case, where the decay of the eigenvalues admits $\lambda_i \propto O(i^{-1})$, then $d_k$ in the worst case is bounded by $O(\frac{1}{\lambda}$, and thus, via Theorem 2, we can deduce that $s_\phi \in O(\sqrt{n}\log{n})$ for the case of $\lambda \in O(\frac{1}{\sqrt{n}}).$
> > >
> > > Note that the choice of $\lambda \in O(\frac{1}{\sqrt{n}})$ is used in the literature of KRR to ensure that the learning rate is $\frac{1}{\sqrt{n}}$ as shown in [2].
> > >
> > > [1] Bach, F. (2017). On the equivalence between kernel quadrature rules and random feature expansions. The Journal of Machine Learning Research, 18(1), 714-751.
> > >
> > > [2] Rudi, A., & Rosasco, L. (2017). Generalization properties of learning with random features. Advances in neural information processing systems, 30.
> > >
> > >
> > >
> > > **Comment 1.2:**
> > >
> > > We have chosen that the size of the distilled set is equal to $s_\phi + 1$ to ensure the existence of $y_S$ with the properties we have sought to maintain. However, this step alone is not enough to ensure that our distilled set admits the provable guarantees stated by Theorem 3. We thus had to also engineer an in-sample prediction function that resides in the same space as the optimal in-sample prediction function which is defined by the solution of the KRR model on the entire data $\mathbf{X}.$
> > >
> > > Note that we don’t ensure the same KRR but rather the same solution for the Ridge regression problem which does not theoretically guarantee the same KRR solution i.e., in sample prediction involving $(S,y_s)$ for $(X,y)$ would be equal to the optimal KRR solution conjured for the $(X,y).$
> > >
> > > Thus the main goal of the other paragraphs is not to show the approximations in the RFF space but to prove the existence of a corresponding set in the original space, with a similar KRR solution in terms of approximation on the whole data P (not ridge regression in the RFF space).
> > >
> > > **Comment 1.3:**
> > >
> > > We thank the reviewer for his keen interest in our paper. We are working now on running an experiment for the multi-class case before the end of the discussion deadline.
> > >
> > > **Comment 2:**
> > >
> > > We appreciate your comment. Indeed, the explanations we provided here were added to the paper itself to improve its clarity. Thus, we greatly appreciate your raised comments and responses.

---

> > > > ### Author Response · Authors · 2023-08-11
> > > > **Thank you + additional answers [Part 2]**
> > > >
> > > >
> > > > **Comment 3:**
> > > >
> > > > Sure! First, we note that theoretically speaking, we provided two bounds as shown in Remark 6. The graph you have mentioned is taking into account the first bound stated in Remark 6, i.e., the one involving the $12\lambda$ term. We checked this one in our graphs as it bound the KRR error, however, we note that the other provided bound on the mean squared prediction errors (see Lines 194-195 in the manuscript) only relies on $8\lambda$ which is much smaller.
> > > >
> > > > To derive the KRR loss bounds (those bounds that are in the provided gaps), we use 3 main steps. The first stage is approximating KRR by Ridge regression in the RFF space, this provides an error of $2\lambda$ (small).
> > > > Then we showed the existence of a small set in the RFF space approximating the Ridge regression loss there on the full data, which results in an approximation of $4\lambda$ (still small).
> > > >
> > > > However, in the last step, we had to link the KRR loss of the distilled set and the KRR loss with respect to the RFF image of the input data. To that end, and in order to separate the terms properly, we had to use the weak triangle inequality (i.e., the version entailing the inequality $|a+b|^2 \leq 2|a|^2 + 2|b|^2$). The idea behind this is to move from the KRR on the RFF space of input data to the KRR on the input data as shown in [LTOS21]. Combining all such steps, we obtained that our bound entails twice the total loss of KRR on the input data in addition to $12\lambda$. This explains why the gap is large.
> > > >
> > > > We note that depending on whether the actual loss of the optimal KRR loss is larger than $12\lambda$, then we can use Remark 6 (specifically the second inequality) and utilize a smaller gap. However, we wanted to remain impartial to whether such a case holds and simply used the bound as is without exploiting the theoretical implications it hold.
> > > >
> > > > Since we are working with the squared loss function (KRR loss function), in order to maneuver from the distilled set to the original set, we had to use the weak triangle inequality in order to use all the inequalities our distilled set admits. This is the main reason why we had the 2-multiplicative factor embedded in our bound. Of course, one can use the other version of the weak triangle inequality that we have proposed, however from a practical point of view, we have seen that the first bound (involving the $12 \lambda$) is tighter.
> > > >
> > > > We believe that our approach raises the first theoretical approach justification of dataset distillation, and we hope that our research will spark new novel approaches in this field, leading to better and tighter theoretical guarantees eliminating some used assumptions and thus getting better practical bounds.
> > > >
> > > > **Comment 4:**
> > > >
> > > > More results both on the multi-class case and for the SVHN dataset are coming soon.

---

> > > > > ### Author Response · Authors · 2023-08-11
> > > > > **Thank you + additional answers [Part 3]**
> > > > >
> > > > > We are glad that we succeed to address part of your concerns! Hoping that this response and the following experimental results that will be added ASAP will address the remaining ones, encouraging you to further increase your score.
> > > > >
> > > > > Please let us know if there is anything else we can improve - so we can make it before the end of the discussion deadline.

---

> > > > > > ### Comment · Reviewer_uCrm · 2023-08-17
> > > > > > **Thanks for the new answer and results + Additional questions**
> > > > > >
> > > > > > I would like to thank the authors for answering my questions and providing additional results on multi-classification and larger datasets. Currently, I still have the following concerns:
> > > > > > 1. I am still curious about the relationship between this work and existing works based on KRR, e.g., [a]. How can existing methods utilize the results of this paper? What is the connection between the random feature space used in [a] and the RFF space in this paper?
> > > > > > 2. I am still confused by the claim in the previous response that "we don’t ensure the same KRR but rather the same solution for the Ridge regression problem which does not theoretically guarantee the same KRR solution". Let's not consider the effect of the regularization term $\lambda$ at first. Indeed, solving the optimal $y_s$ can guarantee the same solution in the RFF space. Why is this not enough in dataset distillation? Let's take the NNGP kernel in [a] as an example. If the solutions in the feature space are consistent, the predictions should also be consistent. Maybe I missed something here but I still didn't get the point here.
> > > > > > 3. After finding the vector of $\alpha$, how to reconstruct the targeting $S$? I noticed that Reviewer BMB1 has a similar concern. But I did not get clues from the authors' responses. Is it related to dual Lagrangian? If so, I think we need to find $\tilde{S}$ first and then use the inverse RFF function to find $S$. If that is the case, discussions on the existence of this inverse function are necessary.
> > > > > >
> > > > > > [a] Efficient Dataset Distillation using Random Feature Approximation (Noel Loo et al., NeurIPS 2022)

---

> > > > > > > ### Author Response · Authors · 2023-08-17
> > > > > > > **Response to additional questions raised by uCrm**
> > > > > > >
> > > > > > > **Question:** I am still curious about the relationship between this work and existing works based on KRR, e.g., [a]. How can existing methods utilize the results of this paper? What is the connection between the random feature space used in [a] and the RFF space in this paper?
> > > > > > >
> > > > > > > **Answer:** We note that the NNGP kernel used in [a] is not a shift-invariant kernel, whereas our theory clearly states that the kernel needs to be shift-invariant to ensure provable guarantees. For references to that this kernel is not shift-invariant, please refer to [2, 3].
> > > > > > > In our work, the theory focused on KIP [1] which is the first result from connecting dataset distillation with KRR. It aims to handle infinite-width neural networks where the NTK of such a network behaves like a shift-invariant kernel [2]. Thus, for the optimization of KIP,, we proved our bound and error.
> > > > > > > For instance, our theory justifies why labelsolve is beneficial in KIP [1]. When future work will suggest further algorithm, for optimzing KIP loss with other techniques, one can check the correctness of their implementations by checking the bound provided in our work. We indeed hope that our work will encourage other work to focus on the theoretical aspect of dataset distillation to yield better bounds and henceforth better algorithms.
> > > > > > >
> > > > > > > [1] Nguyen, T., Novak, R., Xiao, L., & Lee, J. (2021). Dataset distillation with infinitely wide convolutional networks. Advances in Neural Information Processing Systems, 34, 5186-5198.
> > > > > > >
> > > > > > > [2] Jacot, A., Gabriel, F., & Hongler, C. (2018). Neural tangent kernel: Convergence and generalization in neural networks. Advances in neural information processing systems, 31.
> > > > > > >
> > > > > > > [3] Neal, R. M., & Neal, R. M. (1996). Priors for infinite networks. Bayesian learning for neural networks, 29-53.
> > > > > > > ____
> > > > > > > **Question:** I am still confused by the claim in the previous response that "we don’t ensure the same KRR but rather the same solution for the Ridge regression problem which does not theoretically guarantee the same KRR solution". Let's not consider the effect of the regularization term $\lambda$ at first. Indeed, solving the optimal $y_S$ can guarantee the same solution in the RFF space. Why is this not enough in dataset distillation? Let's take the NNGP kernel in [a] as an example. If the solutions in the feature space are consistent, the predictions should also be consistent. Maybe I missed something here but I still didn't get the point here.
> > > > > > >
> > > > > > > **Answer:** For any shift-invariant kernel, we first aim to ensure that $y_S$ will guarantee that the ridge regression in the RFF space involving $(\tilde{S}, y_S)$ is equal to that of the ridge regression involving $(\tilde{X}, y)$. This step alone is not enough, since this does not guarantee that there exists a solution for the KRR problem involving $(S,y_S)$ that would yield our desired approximation since we first need to show that there exists an in-sample function that resides in a ball around the prediction function that the ridge regression would yield when taking $(S,y_S)$ into account.
> > > > > > > This is mainly done since we need to show that there exists a KRR solution involving $(S,y_S)$ and not simply a solution in the RFF space since Lemma 4 indicates that if for any in-sample prediction $f$ there exists $\beta$ (ridge regression solution) that would be utilized to obtain Theorem 2. However, we want to reconstruct $f$ given $\beta$, so in our approach, we have taken the other way around, and thus we still need to prove its existence rather than assume it.
> > > > > > >
> > > > > > > To that end, Lemma 4 was used and we then had to show that such an in-sample function exists when defining such function with respect to $(S, y_S)$. This was done by showing that one could rephrase the equation yielding equation 6 in our manuscript. To ensure the existence of $\alpha$, we had to solve a system of $s_\phi$ equations with $s_\phi + 1$ variables (entries of $\alpha$), where such system of equations would be underdetermined, i.e., have at least one solution (possible $\alpha$).

---

> > > > > > > > ### Author Response · Authors · 2023-08-17
> > > > > > > > **Response to additional questions raised by uCrm [Part 2]**
> > > > > > > >
> > > > > > > > **Question:** After finding the vector of $\alpha$, how to reconstruct the targeting $S$? I noticed that Reviewer BMB1 has a similar concern. But I did not get clues from the authors' responses. Is it related to dual Lagrangian? If so, I think we need to find $\tilde{S}$ first and then use the inverse RFF function to find $S$. If that is the case, discussions on the existence of this inverse function are necessary.
> > > > > > > >
> > > > > > > > **Answer:** We thank the reviewer for raising an interesting question. First note based on your referral to our answer to Reviewer BMB1, we gave an example of one way to construct $S$ – being the corresponding rows in $X$ such that their RFF image is the maximal set of independent rows in $\tilde{X}$. In our latest responses, we highlighted that one can also construct $S$ that in a brute-force manner (searching over sets of $O(s_\phi)$ instance such that their RFF image (i.e., $\tilde{S}$) is either of rank equal to that of $\tilde{X}$ or $s_\phi$. That is the only requirement that ensures the existence of $y_s$.
> > > > > > > >
> > > > > > > >
> > > > > > > > To that end, we discussed the requirements that such an $S$ needs to comply with in order to ensure provable guarantees in the context of dataset distillation.

---

> > > > > > > > > ### Comment · Reviewer_uCrm · 2023-08-20
> > > > > > > > >
> > > > > > > > > Thanks for the reply! Please help me check if my current understanding is correct: The label-solving step is using features in RFF space, while the final goal is on the feature space of infinite-width neural networks, which is related to NTK and KIP. Therefore, the following part is used to build a bridge between the two spaces.
> > > > > > > > >
> > > > > > > > > Plus, there is an additional question: it seems that the construction of $S$ shown by the authors in the response to Reviewer BMB1 is not related to the solved $\alpha$, which results in some gaps here. What is the functionality of $\alpha$, please?

---

> > > > > > > > > > ### Author Response · Authors · 2023-08-20
> > > > > > > > > > **Thanks for your interest and engagement**
> > > > > > > > > >
> > > > > > > > > > Thanks for the reply! Please help me check if my current understanding is correct: The label-solving step is using features in RFF space, while the final goal is on the feature space of infinite-width neural networks, which is related to NTK and KIP. Therefore, the following part is used to build a bridge between the two spaces.
> > > > > > > > > >
> > > > > > > > > > We thank the reviewer for their interest and engagement with us. You are indeed correct.
> > > > > > > > > >
> > > > > > > > > > ______
> > > > > > > > > >      Plus, there is an additional question: it seems that the construction of $S$ shown by the authors in the response to Reviewer BMB1 is not related to the solved $\alpha$, which results in some gaps here. What is the functionality of $\alpha$ , please?
> > > > > > > > > > After finding $S$ such that $\tilde{S}$ either has full rank or rank equal to that of $\tilde{X}$, we aim to find $y_S$ such that the solution in the RFF space involving $(\tilde{S}, y_S)$ is equivalent to that of $(\tilde{X}, y)$. With this in mind, we still need to ensure that there exists a solution in the KRR space (original space, i.e., space of $S$) such that that solution is related in a way to the solution of the $RFF$ space (Lemma 4 in our manuscript). To that end, the construction of $\alpha$ is needed since using this weight vector, we can construct an in-sample prediction function that approximates the prediction quality of the ridge regression model trained on $(\tilde{S}, y_S)$ with respect to the data $(X, y)$.
> > > > > > > > > >
> > > > > > > > > > The idea behind such a step is to mimic the provable guarantees that [LTOS21] ensures – [LTOS21] have shown that one can obtain a provable approximation to the solution of a KRR model by bridging a connection between such a solution and the optimal solution of the ridge regression in the RFF space. Our construction of $\alpha$ aims to ensure the existence of a KRR solution that will ensure the guarantees of Lemma 4 which is primarily the bridge between the two solutions (in the RFF and KRR spaces) that we aim to encapsulate.
> > > > > > > > > >
> > > > > > > > > >
> > > > > > > > > > We are still available to answer any further questions if needed.
> > > > > > > > > > Thank you.

---

> > > > > > > > > > > ### Comment · Reviewer_uCrm · 2023-08-21
> > > > > > > > > > >
> > > > > > > > > > > Thanks for the further clarification and my first question is addressed. For the second, please help me check if my understanding is correct: solving $\alpha$ indicates finding a KRR solution. There are a number of synthetic datasets $S$, maybe an infinite number, that can result in this KRR solution. Your response to Review BMB1 is one way to construct one valid $S$.

---

> > > > > > > > > > > > ### Author Response · Authors · 2023-08-21
> > > > > > > > > > > > **On and $S$ and $\alpha$**
> > > > > > > > > > > >
> > > > > > > > > > > > **Comment:** Thanks for the further clarification and my first question is addressed. For the second, please help me check if my understanding is correct: solving $\alpha$ indicates finding a KRR solution. There are a number of synthetic datasets $S$, maybe an infinite number, that can result in this KRR solution. Your response to Review BMB1 is one way to construct one valid $S$.
> > > > > > > > > > > >
> > > > > > > > > > > >
> > > > > > > > > > > > **Answer:** The reviewer is indeed correct. We aimed to give an easy-to-construct example of $S$ that would result in our reported provable approximation in Theorem 3. In addition, as mentioned before, one can construct $S$ via a brute-force approach to be any set such that $\tilde{S}$ is of full rank (or rank equal to that of $\tilde{X}$). Once $S$ is constructed, $\alpha$ can be computed as it aims to connect between $S$ and $y_S$ such that the resulting in-sample prediction function with respect to $(X, y)$ approximates that of the prediction function in the RFF space with regard to $(X,y)$ obtained by training on $(\tilde{S},y_S)$.
> > > > > > > > > > > >
> > > > > > > > > > > > We are still available to answer any lingering concerns that you might have. Thank you again.

---

> > > > > > > > > > > > > ### Comment · Reviewer_uCrm · 2023-08-21
> > > > > > > > > > > > >
> > > > > > > > > > > > > Thanks the author for the reply! Now most of my concerns are addressed. Indeed, there is a gap between the brute-force approach and the traditional definition of dataset distillation, since most of the samples are directly from the original datasets instead of synthesis, which would result in some gap between theoretical error and actual error. Future works may focus on this. I would like to further increase my score to 5 - borderline accept and encourage the authors to include all the discussions and clarifications in the revision if the paper is accepted.

---

> > > > > > > > > > > > > > ### Author Response · Authors · 2023-08-21
> > > > > > > > > > > > > > **Thank you for the interesting discussion**
> > > > > > > > > > > > > >
> > > > > > > > > > > > > > We truly appreciate your engagement with us during the discussion process. It was indeed beneficial for improving the paper and its clarity!
> > > > > > > > > > > > > > Indeed, we have included all the clarifications and details in our new version.
> > > > > > > > > > > > > >
> > > > > > > > > > > > > > Thanks also for raising your score.

---

### Official Review · Reviewer_piM1 · 2023-07-03

**Soundness:** 3 good
**Presentation:** 3 good
**Contribution:** 2 fair
**Rating:** 6
**Confidence:** 3

**Summary:**

This manuscirpt first give theoretical understanding on synthetic dataset generated in dataset distillation task. In concrete, the authors prove (1) the existance of distilled datasets and (2) the generalization error is related to the "number of effective degrees of freedom" in the random Fourier features (RFF) regime. The theoretical bounds are further verified by simple experiments.


**Strengths:**

1. First theoretical work on the field of dataset distillation and the theoretical results are well established;

2. Theoretically show the correlation between the size of distilled dataset and the characteristics of kernels in RFF regime;

3. Theoretically show the generalization bound w.r.t. distilled datasets in kernel ridge regression (KRR) regime.

**Weaknesses:**

1. The theoretical results are biult on KRR, which has a gap to the finite-width network architectures used in dataset distillation.

2. Can the derived generalization bounds provide insights on developing novel dataset distillation algorithms. For example, imporving the kernel architecture to decrease the right-handed term in the generalization bound to reduce the risk trained on synthetic datasets. In this way, the bound can be seen to be tight and impractical.

**Questions:**

See weekness.

**Limitations:**

See weekness.

---

> ### Author Rebuttal · Authors · 2023-08-08
>
> We extend our appreciation to the reviewer for their expert evaluation, insightful remarks, positive feedback, and valuable suggestions that have contributed to the enhancement of our manuscript.
>
> We now delve into a comprehensive discussion of the concerns raised by the reviewer. We trust that our responses provide a thorough resolution to all your queries, and we look forward to the possibility of you revising your evaluation positively. Should there be any lingering points of concern, we welcome the opportunity to address them to your satisfaction.
>
> **Response to Comment 1.**
>
> We thank the reviewer for pointing this out. Many theoretical papers in the field of deep learning focus on infinite-width neural networks which are easier to analyze and interpret [1,2,3], as well as connecting such networks to other famous models in the field of deep learning [4,5]. In addition, such networks are widely used for the case of practical datasets distillation [6,7]. This phenomenon arises because KRR-based distillation techniques exhibit strong theoretical compatibility with neural networks of infinite width. In this context, the training process of the neural network aligns with kernel regression principles, as elegantly demonstrated by [1].
>
> To this end, we started by analyzing infinite-width architectures that satisfy such helpful theoretical attributes. We believe it is the first step towards a better understanding of dataset distillation, which will allow us to provide better provable distillation algorithms and interpret the theory behind them. We hope that our work will provide the first theoretical stepping stone towards analyzing and better understanding the magic behind dataset distillation techniques (specifically KRR-based).
>
> [1] Jacot, A., Gabriel, F., & Hongler, C. (2018). Neural tangent kernel: Convergence and generalization in neural networks. Advances in neural information processing systems, 31.
> [2] Arora, S., Du, S. S., Hu, W., Li, Z., Salakhutdinov, R. R., & Wang, R. (2019). On exact computation with an infinitely wide neural net. Advances in neural information processing systems, 32.
>
> [3] Yang, G., & Hu, E. J. (2020). Feature learning in infinite-width neural networks.
>
> [4] Lee, J., Bahri, Y., Novak, R., Schoenholz, S. S., Pennington, J., & Sohl-Dickstein, J. (2017). Deep neural networks as gaussian processes.
>
> [5] Sohl-Dickstein, J., Novak, R., Schoenholz, S. S., & Lee, J. (2020). On the infinite width limit of neural networks with a standard parameterization.
>
> [6] Nguyen, T., Novak, R., Xiao, L., & Lee, J. (2021). Dataset distillation with infinitely wide convolutional networks. Advances in Neural Information Processing Systems
>
> [7] Loo, N., Hasani, R., Amini, A., & Rus, D. (2022). Efficient dataset distillation using random feature approximation. Advances in Neural Information Processing Systems
>
>
> **Response to Comment 2.**
>
> This is indeed important, and we thank the reviewer for pointing this out. For space limit purposes, we kindly refer the reviewer to our answer to Question 2 of Reviewer myfC. For your convenience: https://openreview.net/forum?id=XWYv4BNShP&noteId=q8bbHghfO8.

---

### Official Review · Reviewer_BMB1 · 2023-07-06

**Soundness:** 2 fair
**Presentation:** 1 poor
**Contribution:** 2 fair
**Rating:** 4
**Confidence:** 2

**Summary:**

The paper attempts to provide the first theoretical guarantees on the existence of dataset distillation, under the setup of kernel ridge regression. The proof techniques are mainly based on theory of random fourier features. They also provide experiments which are indicated to support their theoretical results.

**Strengths:**

1. The paper is the first attempt to theoretically guarantees the existence of dataset distillation, which is an important topic for efficient learning.
2. Experimental results are seemingly consistent with the theoretical results, which could strength their claim if the relationship between the theoretical results and the experimental protocol was clear.

**Weaknesses:**

1. The paper seems not to pay attention to readability of its theoretical results. E.g., Theorem 2 is explained by just 2 lines (l.130-131) without any proof; the notations and statements in Theorem 2 and Theorem 3 are not sophisticated.
2. The proof of Theorem 2 is not provided even in the supplementary materials, but just stated as "A result of the proof of Theorem 1 and Corollary 2 of [ LTOS21]".
3. I cannot find the proof of the existence of the distilled dataset S in the proof of Theorem 3, which is the main claim in this paper.
4. The relationship between the theoretical construction (which I cannot find) and the distillation method used in experiments is unclear. So I'm not confident in whether the experiments really support their theoretical results.

**Questions:**

1. Where is the existence of S proved?
2. Can you give the intuition or short strategy of the proof before Theorem3? For example, why can the number of the distilled data S be $s_\phi + 1$? How can we construct it explicitly?
3. Where did you explain what distillation algorithm is used in experiments? How does the algorithm relate to the construction in the theoretical results?

**Limitations:**

Limitation is discussed in the last section.

---

> ### Author Rebuttal · Authors · 2023-08-08
>
> We wish to extend our heartfelt appreciation to the esteemed reviewer for their dedicated commitment to meticulously evaluating our paper. The thoughtful points and careful reading hold a pivotal role in the refinement of our work. We have diligently addressed each of these valuable concerns, and we remain enthusiastic about engaging in further dialogues with the reviewer to ensure the complete resolution of any lingering matters.
>
> Summing up, we hold the belief that our comprehensive response will ideally have a positive effect on your assessment, potentially boosting your score.
>
> **Response to Comment 1:**
>
> Thank you for raising this comment. This is indeed important. We kindly refer the reviewer to the section “Clarity of our theoretical results” in the general comments  that provides most of the added details to our manuscript -- explaining theorems 2 and 3
>
> For your convenience: https://openreview.net/forum?id=XWYv4BNShP&noteId=RA4SAIBzsG
>
> **Response to Comment 2:**
>
> We thank the reviewer for the careful reading and professional review.  The proof of Theorem 2 can be done by following some derivations from the proofs of Theorem 1 and Corollary 2 from [LTOS21]. Following your insightful comment, and for better readability and completeness, we have restated these derivations in the appendix of the paper, while pointing out the original derivations from [LTOS21].
>
> **Response to Comment 3:**
>
> We thank the reviewer for raising this concern. Let $X$ be the input data, $S$ be the desired set we wish to prove its existence and denote by $\tilde{\mathbf{S}}$ and $\tilde{\mathbf{X}}$ the matrix corresponding to the instances of the image of instances of $S$ in the space of RFF, and the matrix corresponding to the instances of the image of instances in $X$ in the RFF space, respectively.
> First, note that $S$ can be any set of points as long as the labels $y_S$ satisfy that:
>  * The solution of the ridge regression on $\left( \tilde{\mathbf{S}}, y_S \right)$ is equivalent to the solution of the ridge regression on $\left(\tilde{\mathbf{X}}, y \right)$ (the image of input data $X$ in the space of RFF); see lines 151-155 in our manuscript.
>
> In other words, we aim to ensure that the optimal solution in the context of Ridge regression on the RFF image of $X$ and its corresponding labels $y$ is identical to the optimal solution in the context of Ridge regression on the RFF image of $S$ and its corresponding labels $y_{S}$; this was done at lines 151–155.
>
> The motivation behind such a goal lies in the core of Lemma 4 which indicates that:
>  * For every KRR in-sample prediction function $f$ with respect to $X$ defined by the instances of $S$ (referred to $f_{\mathbf{S}}\left[ \mathbf{X} \right]$ in our context), there exists a Ridge regression solution $\beta$ obtained from training a Ridge regression model on $\left( \tilde{S}, y_S \right)$ that admits an additive approximate to the MSE of $f$ and $\mathbf{X}\beta$; see the summation term in the inequality at Lemma 4.
>
> With this in mind, we aimed at constructing such an in-sample prediction function using a Ridge regression solution. To that end, to ensure proper usage of Lemma 4, we have shown that through equation reformulation (involving $\beta$ which is the solution of the aforementioned Ridge regression) and a system of equation solving, there exists a KRR in-sample prediction with respect to the input data involving the distilled set $S$, which satisfies Lemma 4 with respect to $\beta$. With this in mind and the weak triangle inequality (Lemma 5), we derive (i) and (ii) of Theorem 3. This concludes the existence of $S$ that depends on a certain structure that the labels of the distilled set need to admit.
>
> Thus, in summary, by showing that (i) $S$ can be any set with (ii) its labels satisfying a concrete structural property, and the specific (iii) derivation of the KRR solution on $S$, we proved Theorem 3.
>
> All of these details were added to the appendix and a part of them in the relevant places of the proof of Theorem 3 following your fruitful comment.
>
> **Response to Comment 4:**
>
> We apologize for missing this. In our experiments, we were directly minimizing the left-hand side of the equation in line 205 which directly corresponds to the KIP [1]  loss which uses KRR for distillation. This method directly satisfies our theoretical assumptions and guarantees.
>
>     [1] Nguyen, T., Novak, R., Xiao, L., & Lee, J. (2021). Dataset distillation with infinitely wide convolutional networks. Advances in Neural Information Processing Systems, 34, 5186-5198.
>
> ------
>
> **Answer to Question 1:**
>
> Please see the answer to Comment 3.
> For instance, see also Lines 151-155 for the construction of $y_S$ given $S$, and the derivation of $f_{\mathbf{S}}[X]$ at lines 169-187.
>
> **Answer to Question 2:**
>
> Sure and thank you for pointing this out. We wrote a detailed explanation of the proof of Theorem 3 in the answer to Comment 1. Also, see the response to Comment 3 regarding the construction of S.
> Regarding the use of $s_{\phi} + 1$: This is done to ensure that the equation system that led to the derivation of $y_S$ has infinite solutions which leads to the freedom of choosing any $S$ to be a distilled set as long as $y_S$ maintains a connection to the Ridge regression solution in the RFF space as elaborately explained in Comment 3.
>
> **Answer to Question 3:**
>
> We apologize for missing this. In our experiments, we were directly minimizing the left-hand side of the equation in line 205 which directly corresponds to the KIP [1] loss which uses KRR for distillation. This method directly satisfies our theoretical assumptions and guarantees.

---

> > ### Comment · Reviewer_BMB1 · 2023-08-12
> > **Thank you for the rebuttal**
> >
> > > Thus, in summary, by showing that (i) $S$ can be any set with (ii) its labels satisfying a concrete structural property, and the specific (iii) derivation of the KRR solution on $S$, we proved Theorem 3.
> >
> > My question remains unsolved: how you constructed or proved the existence of $S$ satisfying these requirements? Could you clarify this? I'm not sure, so sorry if I missed it.
> >
> > > We apologize for missing this. In our experiments, we were directly minimizing the left-hand side of the equation in line 205 which directly corresponds to the KIP [1] loss which uses KRR for distillation. This method directly satisfies our theoretical assumptions and guarantees.
> >
> > Did you explain this somewhere in the main paper? Sorry if I missed it.

---

> > > ### Author Response · Authors · 2023-08-12
> > > **Thank you + additional clarifications**
> > >
> > > We express our gratitude to the reviewer for their prompt feedback and active participation in the discussion. We believe this is highly beneficial for improving our paper and its clarity.
> > >
> > > ____
> > > **Regarding the existence of $S$**
> > >
> > > We apologize if the previous answer was not clear. To make sure we cover all the relevant details, we first provide a set of steps to generate the distilled set $(S, y_S)$, then, we explain “why” following these steps guarantees that the required characteristics hold on the generated set $(S, y_S$). Here are the steps to generate $S$:
> > >
> > >  1. $S \gets $ Sample a set of $s_\phi + 1$ instances from the input space uniformly at random.
> > >  2. Let $\tilde{S}$ be the RFF image of $S$ and let $\tilde{X}$ be the RFF image of the input data $X$.
> > >  3. Let $y_S$ be the labels of $S$ defined as a solution to the following equality (such a solution indeed exists, since we have $s_\phi + 1$ variables, and $s_\phi$ equations):
> > > $$\left( \tilde{S}^T \tilde{S}  + \lambda n s_\phi \lambda I_{s_\phi}\right) \left( \tilde{X}^T \tilde{X}  + \lambda n s_\phi \lambda I_{s_\phi}\right)^{-1}y = \tilde{S}^Ty_S$$
> > >  4. Let $\beta$ be be the solution of the Ridge regression problem involving $(S, y_S)$:
> > > $$ \beta \gets \left( \tilde{S}^T \tilde{S}  + \lambda n s_\phi \lambda I_{s_\phi}\right)^{-1} \tilde{S}^Ty_S$$.
> > >  5. Following Lemma 4, find an in-sample prediction function $f_{S}[X]$ such that
> > > $$\frac{1}{n} \sum\limits_{i=1}^n \left| f_S(X_{i*}) - \tilde{X}_{i*}\beta \right|^2 \leq 2\lambda$$
> > >
> > > *On step 3.* Here, we generate the set of labels $y_S$ ensuring that the ridge regression solution with respect to the RFF image of the distilled set $(S,y_S)$ is identical to the ridge regression solution with respect to the RFF image of the $(X,y)$; such a solution is referred to as $\beta$ above and also throughout our manuscript (Step 4). The intuition behind this step is to leverage the use of Theorem 2 in our context.
> > >
> > > *On step 5.* Lemma 4 in our manuscript enables us to move from in-sample prediction defined over the distilled set with respect to the input data $X$ to the solution $\beta$ which is of high importance in our derivations. To this end, step 5 aims to ensure that there exists such an in-sample prediction function and we have ensured its existence via variable reformulation and equation-solving techniques as elaborated in lines 162–187 of our manuscript.
> > >
> > > To obtain the provable guarantees associated with our distilled set $(S,y_S)$, we used the weak triangle inequality to split $\frac{1}{n} \sum\limits_{i=1}^n \left| y_i - f^\lambda_S(X_{i*})\right|^2$ to a linear combination of
> > >
> > >  (i) $\frac{1}{n} \sum\limits_{i=1}^n \left| y_i - f^\lambda_{\left[\tilde{\mathbf{X}}, y, \phi\right]}(X_{i*}) \right|^2$, and
> > >
> > >  (ii)  $\frac{1}{n} \sum\limits_{i=1}^n \left|  f^\lambda_S(X_{i*}) -  f^\lambda_{\left[\tilde{\mathbf{X}}, y, \phi\right]}(X_{i*}) \right|^2$.
> > >
> > > Observe that by construction, it holds that  $f^\lambda_{\left[\tilde{\mathbf{X}}, y, \phi\right]}(X_{i*}) = X_{i*}  \left( \tilde{X}^T \tilde{X} + \lambda n s_\phi \lambda I_{s_\phi}\right)^{-1}y = X_{i*}\beta$ for every $i \in [n]$. Thus, we note that (i) is bounded by Theorem 2 whereas (ii) is bounded using Lemma 4 and by the construction of our in-sample prediction function $f_S[X]$.
> > >
> > >
> > > **Regarding the minimized loss in our experiments**
> > >
> > > Indeed, we missed clarifying these details regarding the generated set of points in the text itself; we minimize the left-hand side of the equation below line 205, i.e., $\frac{1}{n} \sum\limits_{i=1}^n \left| y_i - f^\lambda_S(X_{i*})\right|^2$.
> > >
> > > Following your comments, these details were added to the paper. Thanks for the careful reading.
> > >
> > > _______
> > >
> > > We hope that our responses above effectively address your concerns, and motivate you to consider raising your score. Should you have any further areas of improvement in mind, please don't hesitate to inform us. Your feedback would be immensely valuable as we strive to make necessary enhancements before the conclusion of the discussion period.

---

> > > > ### Comment · Reviewer_BMB1 · 2023-08-13
> > > >
> > > > Thank you for the additional details on the proof. Unfortunately, I still have some (possibly critical) concerns about it.
> > > >
> > > > > 1. $S \leftarrow$ Sample a set of $s_\phi +1$ instances from the input space uniformly at random.
> > > >
> > > > The argument that $S$ is taken to be a uniform random matrix was not provided in the original proof. Also the uniform randomness was not used in the argument in the reply.
> > > >
> > > > > Let $y_S$ be the labels of defined as a solution to the following equality (such a solution indeed exists, since we have variables $s_{\phi}+1$, and equations $s_{\phi}$)
> > > >
> > > > To guarantee the existence of such a solution, counting the numbers of variables and equations is not enough in general. For example, the claim does not hold in the trivial case $\widetilde{S}=0$. Thus they have to guarantee the invertiblity of $\widetilde{S}$, which should involve some functional property of the RFF mapping for the uniformly sampled input $S$. However, there is no discussion on such properties. At least, for such inverse to exist, the dimension $s_\phi$ must be lower than or equal to $n$ but even this is non-trivial from Theorem 2.
> > > >
> > > > Overall, I strongly suggest to put more effort into polishing their writing so that readers can understand the theoretical results and the experimental setting more intuitively and clearly.

---

> > > > > ### Author Response · Authors · 2023-08-13
> > > > > **Response to Official Comment by Reviewer BMB1**
> > > > >
> > > > > **Regarding $S$ in the pseudo-code:**
> > > > >
> > > > > We wrote “uniformly at random” just for the simplicity of the explanation in the rebuttal, i.e., to make the pseudocode look easier to understand. However, in the proof of the paper, the original rebuttal, and now, we state clearly that it is not needed.
> > > > > We can use any set $S$ as long as the rank of its RFF image (i.e., $\tilde{S}$) is full (i.e., $s_\phi$); We added these details to Line 150.
> > > > >
> > > > > This indeed emerges from our proof of Theorem 3. Specifically, the existence of $y_S$ entails that $\left( \tilde{S}^T \tilde{S}  + \lambda n s_\phi \lambda I_{s_\phi}\right)$ is of full rank, as well as $\tilde{S}$ is also of full rank (i.e., $s_\phi$). Otherwise, the Ridge regression solution involving $(S, y_S)$ cannot be identical to that of the Ridge regression solution on $(\tilde{X}, y)$ which is a pivotal step in our proofs.
> > > > >
> > > > > Throughout the paper, we did mean any set $S$ with the above trait, i.e., $\tilde{S}$ is of full rank. Hence, $S$ can be generated as you wish: e.g., by sampling, or by defining it manually as long as the rank in the RFF space is full. Recall that we do not suggest an algorithm for how to compute $S$, we only show the existence of $S$.
> > > > >
> > > > > Thank you for sharing your concern regarding the clarity of the response. We have updated the manuscript accordingly. To this end, your comment has filled in this missing detail. We hope that now it is clearer.
> > > > >
> > > > >  ___
> > > > > **Regarding the second comment**
> > > > > We thank the reviewer for pointing this out. Indeed the property that should hold is that the matrix in the RFF space ($\tilde{S}$) has to be full rank. We added these details to Line 150.  Since we do not suggest an algorithm to create such a set. Thus we can use any set $S$ of size $s_\phi+1$ satisfying that its RFF mapping yields a full rank matrix.
> > > > >
> > > > > **Regarding $s_\phi$ and its connection to $n$**
> > > > >
> > > > > We kindly refer you to our response to *reviewer uCrm* at https://openreview.net/forum?id=XWYv4BNShP&noteId=rLEUwOf9V6
> > > > >
> > > > > For your convenience, we restate it here -- To better quantify $s_\phi$ which in turn determines the size of the distilled set, we need to measure $d_K$ which is the trace of $\mathbf{K} \left( \mathbf{K} + n\lambda I_n\right)^{-1}$:
> > > > >  1. For the case where $K$ has a finite rank, i.e., the number of positive eigenvalues is lower than $n$, then $s_\phi \in \Omega(1)$.
> > > > >  2. As for the exponential decay, it occurs when the kernel is Gaussian and the marginal distribution of the input data (e.g., images) is sub-Gaussian. In such a case, it was shown in [1] that $d_K \in O\left(\log{\frac{1}{\lambda}} \right)$. Thus $s_\phi$ is poly-logarithmic in $n$ when $\lambda := O\left( \frac{1}{\sqrt{n}}\right)$.
> > > > >  3. For the case where the Hilbert space $\mathcal{H}$ is also a Sobolov space of order $\gamma$ larger or equal to 1, then $d_K \in O\left(\frac{1}{\lambda^{2\gamma}}\right)$ which in the case of $\lambda := O\left( \frac{1}{\sqrt{n}}\right)$, we have $s_\phi \in \Omega\left(n^{\frac{1}{4\gamma}}\right)$.
> > > > >  4. In the general case, where the decay of the eigenvalues admits $\lambda_i \propto O(i^{-1})$, then $d_k$ in the worst case is bounded by $O(\frac{1}{\lambda}$, and thus, via Theorem 2, we can deduce that $s_\phi \in O(\sqrt{n}\log{n})$ for the case of $\lambda \in O(\frac{1}{\sqrt{n}})$.
> > > > >
> > > > > Note that the choice of $\lambda \in O(\frac{1}{\sqrt{n}})$ is used in the literature of KRR to ensure that the learning rate is $\frac{1}{\sqrt{n}}$ as shown in [2].
> > > > >
> > > > >   [1] Bach, F. (2017). On the equivalence between kernel quadrature rules and random feature expansions. The Journal of Machine Learning Research, 18(1), 714-751.
> > > > >
> > > > >   [2] Rudi, A., & Rosasco, L. (2017). Generalization properties of learning with random features. Advances in neural information processing systems, 30.
> > > > >
> > > > > ___
> > > > >
> > > > >     I strongly suggest to put more effort into polishing their writing
> > > > > Indeed! We have revised a lot of the wiring as we shared in the main rebuttal response: We have been improving the writing following all of the reviewers' comments, and we keep doing it for every new comment. This is the power of the OpenReview process, and we hope to make the most of this process.
> > > > >
> > > > >
> > > > > Thank you again for engaging with us during the discussion process.

---

> > > > > > ### Comment · Reviewer_BMB1 · 2023-08-13
> > > > > >
> > > > > > > Throughout the paper, we did mean any set $S$ with the above trait, i.e., $\widetilde{S}$ is of full rank. Hence, $S$ can be generated as you wish: e.g., by sampling, or by defining it manually as long as the rank in the RFF space is full.
> > > > > >
> > > > > > What I have been asking from the beginning is: Please directly prove the full rankness of such $\widetilde{S}$ for any $S$ you like. The author seems to avoid this argument throughout their rebuttal and initial submission.
> > > > > >
> > > > > > > For your convenience, we restate it here -- ...
> > > > > >
> > > > > > Please answer to my question directly if you want to make a rebuttal.

---

> > > > > > > ### Author Response · Authors · 2023-08-14
> > > > > > > **Response to Official Comment by Reviewer BMB1**
> > > > > > >
> > > > > > >          The author seems to avoid this argument throughout their rebuttal and initial submission.
> > > > > > >
> > > > > > > We respectfully disagree with such raised comment. We are definitely not avoiding (nor trying to) this argument. It's possible that we might not have fully understood your question, or perhaps your question wasn't communicated with absolute clarity. We hope that the following responses meet your satisfaction. Please let us know.
> > > > > > > ____
> > > > > > >        What I have been asking from the beginning is: Please directly prove the full rankness of such $\tilde{S}$ for any $S$ you like.
> > > > > > > We give an example for constructing $S$ such that $\tilde{S}$ would be of full rank ($s_\phi$). Let $\tilde{X} \in \mathbb{R}^{n \times s_\phi}$ be the RFF image of $X$:
> > > > > > >
> > > > > > > * If $\tilde{X}$ is of full rank (the rank is $s_\phi$), then there exists a subset $S$ of the rows of $X$ such that its image in the RFF space is a linearly independent subset $\tilde{S}$ of the rows of $\tilde{X}$. Such a subset is of full rank as it is linearly independent. Notes:
> > > > > > >    * If you also want an example of how to find $\tilde{S}$ – Such an algorithm entails the use of QR decomposition; please see https://www.mathworks.com/matlabcentral/fileexchange/77437-extract-linearly-independent-subset-of-matrix-columns for the implementation of such an algorithm.
> > > > > > >    * Also note that we do not force to use this specific subset, but this is an example of one as you requested (as requested by you: ”for any set you like”).
> > > > > > >
> > > > > > >
> > > > > > >  * If $\tilde{X}$ is rank deficient, i.e., $r < s_\phi$ where $r$ denotes the rank of $\tilde{X}$, then the problem by it self requires $r$ independent vectors, i.e, one can represent $\left( \tilde{X}^T \tilde{X}  + \lambda n s_\phi \lambda I_{s_\phi}\right)^{-1} \tilde{X}^T y$ by $A z$ where $A \in \mathbb{s_\phi \times r}$ and $z \in \mathbb{R}^r$. This is since the rank of $\left( \tilde{X}^T \tilde{X}  + \lambda n s_\phi \lambda I_{s_\phi}\right)^{-1} \tilde{X}^T$ is at max the minimum between the rank of $\left( \tilde{X}^T \tilde{X}  + \lambda n s_\phi \lambda I_{s_\phi}\right)^{-1}$ and the rank of $\tilde{X}^T$. Such rank is bounded from above by $r$ since the rank of $\left( \tilde{X}^T \tilde{X}  + \lambda n s_\phi \lambda I_{s_\phi}\right)^{-1}$ is $s_\phi$.
> > > > > > >
> > > > > > > To this end, we can invoke the first case with respect to $(B,z)$. Thus, by choosing $S$ to contain the $r$ linearly independent rows in $\tilde{X}$ and a additional row from $\tilde{X}$ (not already in $S$), we can solve the equation below for $y_S$:
> > > > > > >
> > > > > > > $$\left( \tilde{S}^T \tilde{S}  + \lambda n s_\phi \lambda I_{s_\phi}\right) Bz  = \tilde{S}^T y_S$$.
> > > > > > >
> > > > > > > In addition, for your convenience, we wrote an example (in MATLAB) practically implementing the above.
> > > > > > >
> > > > > > >     function test(n, d)
> > > > > > > 		for i = 1 : 10
> > > > > > > 	    	A = randn(n,d);
> > > > > > > 	        [U, D, V] = svd(A, 0);
> > > > > > > 	        r = randi([2,round(d/5)], 1);
> > > > > > > 	        D(r:end, :) = 0;
> > > > > > > 	        B = U * D * V';
> > > > > > > 	        y = rand(n, 1);
> > > > > > > 	        beta = inv(B' * B + log(d) * eye(d)) * B' * y;
> > > > > > > 	        [C, ~] = licols(B'); % find linearly independent rows in B
> > > > > > > 	        C = C'; % Ensuring rows of C are rows of B
> > > > > > > 	        z = mldivide(C', ((C' * C + log(d) * eye(d)) * beta));
> > > > > > > 	        assert(norm(inv(C' * C + log(d) * eye(d)) * C'*z - beta) < 1e-12);
> > > > > > > 	    end
> > > > > > > 	end
> > > > > > >
> > > > > > > Note that lcols is from [here](https://www.mathworks.com/matlabcentral/fileexchange/77437-extract-linearly-independent-subset-of-matrix-columns).
> > > > > > >
> > > > > > >
> > > > > > > In summary, we let $S$ be the rows in $X$ that are associated with indices of the linearly independent rows in $\tilde{X}$ and we also add any row in $X$ to the set $S$ (which is not contained in $S$) to ensure that the number of instances in $S$ is $s_\phi + 1$ (or equivently $r+1$ ).
> > > > > > > _____
> > > > > > >     Please answer to my question directly if you want to make a rebuttal.
> > > > > > > In the previous response, we have answered your question regarding the quantity $s_\phi$ and its connection to $n$:
> > > > > > >
> > > > > > > ``the dimension $s_\phi$ must be lower than or equal to $n$ but even this is non-trivial from Theorem 2.’’
> > > > > > >
> > > > > > > Since we already answered such a question to Reviewer uCrm who raised it earlier, we did mention such an answer and restated it for your convenience.

---

> > > > > > > > ### Comment · Reviewer_BMB1 · 2023-08-17
> > > > > > > >
> > > > > > > > > Since we already answered such a question to Reviewer uCrm who raised it earlier, we did mention such an answer and restated it for your convenience.
> > > > > > > >
> > > > > > > > My question was Question 1, which should be definitely a clear question. The author was repeating the same discussion as in the main paper and did not provide any explanation for the existence, at least until now.
> > > > > > > >
> > > > > > > > > We give an example for constructing $S$ such that $\widetilde{S}$ would be of full rank ...
> > > > > > > >
> > > > > > > > Thanks for the additional proof. I wonder why the author did not provide such a valid argument from the beginning or in their initial submission. Anyway, the argument seems valid and indeed critical because it also shows the construction of $S$. I guess that the magic number $s_\phi+1$, which was also questioned by Reviewer BMB1, would be totally needless if S is taken as the above construction. On the other hand, I also notice that this is not distillation, but just data selection/pruning. The author claims that $S$ can be arbitrary if satisfying their requirements, but they did not provide any theoretical evidence beyond data selection as above, i.e., **whether or not distillation is superior to just data selection in terms of the number of distilled samples $S$** for example. Also, in their response, they initially claimed that $S$ can be constructed by uniform sampling, but the above argument is not for that case. It should be compared to the theoretical results in previous literatures on data selection or pruning. I strongly suggest the author (i) to rearrange and clarify their proof at least by including the above argument for the existence of S, (ii) to make more effort to improve their mathematical/logical writing so that the reader can easily find what is proved or not in this paper (at least the author themselves did not notice that the existence of S was not proved) and (iii) to add more discussions in relation to data selection/pruning.

---

> > > > > > > > > ### Author Response · Authors · 2023-08-17
> > > > > > > > > **Response to Official Comment by Reviewer BMB1**
> > > > > > > > >
> > > > > > > > > Thanks for the additional proof. I wonder why the author did not provide such a valid argument from the beginning or in their initial submission.
> > > > > > > > >
> > > > > > > > > In our initial submission, the proof was assumed for any $S$ such that $\tilde{S}$ is of full rank. Now, we have added all the relevant details to remove such innate assumptions, clarifying and giving an example of such $S$. We indeed did not hide such details.
> > > > > > > > > ____
> > > > > > > > >     I guess that the magic number $s_\phi + 1$, which was also questioned by Reviewer BMB1, would be totally needless if S is taken as the above construction.
> > > > > > > > > Do you mean the “+1”? If so, we have removed it:
> > > > > > > > >  * Observe that if it holds for $s_\phi$ then it will also hold for $s_\phi +1$.
> > > > > > > > >  * In general, it is common to use the $O$ notation in theoretical papers to cover all hidden constants. However, in our responses and in the paper, we aimed to be as specific as possible.
> > > > > > > > > ____
> > > > > > > > >      On the other hand, I also notice that this is not distillation, but just data selection/pruning.   The author claims that $S$ can be arbitrary if satisfying their requirements, but they did not provide any theoretical evidence beyond data selection as above, i.e., whether or not distillation is superior to just data selection in terms of the number of distilled samples for example.  Also, in their response, they initially claimed that can be constructed by uniform sampling, but the above argument is not for that case.  It should be compared to the theoretical results in previous literatures on data selection or pruning.
> > > > > > > > >
> > > > > > > > > Indeed, subset selection techniques can be thought of as a special case of distillation techniques since distillation techniques aim to find a general set that approximates the input data with respect to a given model.
> > > > > > > > > However, note that our theoretical approach does not guarantee that the labels of our distilled set are inside the set of labels in the input data. On the other hand, data subset selection techniques do not alter the labels but rather subsample from the set of labels associated with the input data. Hence, it is not exactly subset selection.
> > > > > > > > > In practice, it is well known that distillation techniques achieve better results than subset selection. However, in theory, to the best of our knowledge, we are the first to provide such bounds.
> > > > > > > > >
> > > > > > > > > Finally, we gave such an example to emphasize that one can construct a set that is not necessarily a subset of the data. To that end, one can also brute-force on subsets in the input space such that the rank of their RFF image is either the rank of $\tilde{X}$ or $s_\phi$ and such a set can be regarded as $S$.
> > > > > > > > >
> > > > > > > > > ____
> > > > > > > > >     I strongly suggest the author (i) to rearrange and clarify their proof at least by including the above argument for the existence of S, (ii) to make more effort to improve their mathematical/logical writing so that the reader can easily find what is proved or not in this paper (at least the author themselves did not notice that the existence of S was not proved) and (iii) to add more discussions in relation to data selection/pruning.
> > > > > > > > > We have updated the manuscript accordingly:
> > > > > > > > >  * We added examples of constructing $S$.
> > > > > > > > >  * We have incorporated our fruitful discussion into our paper.
> > > > > > > > >  * We have also discussed the relation to data selection pruning explaining that our result holds for subset of the data with different labels (not necessarily from the set of input labels) which can be thought of as a hybrid between distillation and subset selection.

---

### Official Review · Reviewer_mfyC · 2023-07-24

**Soundness:** 3 good
**Presentation:** 3 good
**Contribution:** 3 good
**Rating:** 5
**Confidence:** 2

**Summary:**

This paper presents a theoretical analysis of dataset distillation, specifically focusing on the size and approximation error of distilled datasets. The authors provide bounds on the sufficient size and relative error of distilled datasets for kernel ridge regression (KRR) based methods using shift-invariant kernels. They prove the existence of small distilled datasets and show that a KRR solution can be generated using these distilled datasets that approximate the solution obtained on the full input data. The theoretical results are validated through empirical experiments on synthetic and real datasets.

**Strengths:**

(1) The paper addresses an important problem in dataset distillation by providing theoretical bounds on the size and approximation error of distilled datasets. This fills a gap in the literature where previous work has mainly been empirical.
(2) The use of random Fourier features (RFF) and kernel ridge regression (KRR) provides a solid theoretical foundation for the analysis.
(3) The paper includes both analytical proofs and empirical validation to support the theoretical results.

**Weaknesses:**

(1) The evaluation of the proposed method could be further strengthened by comparing it with other baseline methods in the field of dataset distillation.
(2) The clarity of the exposition could be improved, as some sections of the paper are not easy to understand without prior knowledge of the topic.
(3) In line 223, two clusters and each cluster has 5000 points. Will they conflict with 10^5 points?
(4)The paper should provide more experiments on real and bigger datasets and the visualization of KRR predictive functions on the MNIST dataset.

**Questions:**

(1) Can you provide more experimental results on larger real-world datasets and make appropriate visualizations to prove effectiveness?
(2) Can you provide more explanation about how understanding data distillation can guide the performance of data distillation?

**Limitations:**

Not fully discussed how a better understanding of data distillation will guide and evaluate subsequent data distillation work.

---

> ### Author Rebuttal · Authors · 2023-08-08
>
> The reviewer's comments and expert evaluation are highly valued by us. Indeed, incorporating your feedback has already led to enhancements to the paper. We eagerly anticipate continued interaction with the reviewer during the forthcoming open discussion phase.
> We have meticulously addressed each of the comments and questions raised in the initial review. We remain hopeful that, based on our comprehensive responses, the reviewer might contemplate raising the score. Should there be a need for additional clarification, please feel free to contact us without hesitation.
>
> **Response to Comment 1 and Comment 4:**
>
> Thanks for pointing this out. We first want to clarify that in this work, we did not propose new (competing) methods for distillation. We aimed at providing the first proof that answers the questions "Why do KRR-based distillation methods work?” and “Which guarantees about the size and error can we obtain?" Notably, this paper is the first to prove the existence of a small distilled set with its approximation error in the field of data distillation. The experiments were basically conducted to practically validate the theoretical analysis.
>
> However, following your valuable comment and to further improve the practical justification of our work, we have added more experiments on Cifar10 (see the attached PDF) and we are running now more experiments on SVHN -- will be added once done.
>
> For visualizing KRR predictive functions on the MNIST dataset, can you please elaborate more on that? We will be happy to do so.
> Finally, our paper's objective is to establish the achievable loss bounds using the KIP dataset distillation algorithm. Given this, we do not see how we should benchmark other methods, as our derived bounds aren't pertinent to their evaluation.
>
> **Response to Comment 2:**
>
> This is indeed important! Following your comment, we have revised the writing of the paper, by providing more details before each theorem/claim/definition. We also added an intuition paragraph behind the proof idea; please see the "Clarity of our theoretical results" section above that provides most of the added details to our manuscript -- explaining theorems 2 and 3.
>
> For your convenience: https://openreview.net/forum?id=XWYv4BNShP&noteId=RA4SAIBzsG
>
> **Response to Comment 3:**
>
> Thanks for the careful reading. We applied thresholding to the data points so they wouldn't conflict as depicted in Figure 1. This would ensure that the KRR on this data is able to distinguish between the two classes. Following this comment, we have clarified that in the paper.
>
> ------
> **Answer to Question 1:**
>
> Certainly, see responses to comments 1 and 4.
>
> **Answer to Question 2:**
>
> Certainly, and thanks for raising this. We first state that in general, deriving bounds on the size of the distilled sets helps researchers develop new distillation algorithms and test them on new datasets. Such bounds on the size and approximation error provide a mechanism to debug new suggested algorithms on these new datasets. Additionally, understanding the theory behind the existence of such small distilled sets (with providing proofs) is the first stepping stone towards providing provable dataset distillation techniques which do not exist at all in the literature. In this case, the final (long-term) goal is to leverage this knowledge and understanding of dataset distillation and provide algorithms that provably generate a small set $S$ that encapsulates all of the information in the input data $X$, and thus will guarantee the success of the training of a deep learning model on the distilled data.
>
> We believe that our paper is the first step towards a better understanding of dataset distillation, which will allow the research community to provide better provable distillation algorithms and interpret the theory behind them. We hope that our work will provide the first theoretical stepping stone towards analyzing and better understanding the magic behind dataset distillation techniques (specifically KRR-based).
>
> Furthermore, specifically in our work, we note that our theoretical derivations indicate that any set can be distilled (achieving the approximation error we provided) as long as the labels satisfy that the solution of the ridge regression which involves the mapped distilled set via random Fourier features (RFF) is equivalent to that of the solution of the ridge regression which involves the mapped input data via random Fourier features (RFF) — From a practical point of view, this serves to indicate the success of using LabelSolve (LS) [2, 3] which aims to learn the labels given a set of distilled points, which minimizes the KRR error between the distilled set and input data, and to that end, our paper can be regarded also as a theoretical justification for such a method. We also note that our method can be used to guide a distillation algorithm in finding the best labels given the distilled set, i.e., one can use our theoretical derivations to guide a distillation method in refining the labels of the distilled set, allowing for provable guarantees and better results in the case of KRR-based distillation techniques, which then one can apply bi-level based optimization which can combine both LabelSolve for instance and KIP (or RFAD [1]) using our theory to better direct these optimizers (instances and their labels) to a better-distilled set. We leave this as an open question and direction in the field of dataset distillation.
>
>
>     [1] Loo, N., Hasani, R., Amini, A., & Rus, D. (2022). Efficient dataset distillation using random feature approximation. Advances in Neural Information Processing Systems, 35, 13877-13891.
>
>     [2] Nguyen, T., Novak, R., Xiao, L., & Lee, J. (2021). Dataset distillation with infinitely wide convolutional networks. Advances in Neural Information Processing Systems, 34, 5186-5198.
>
>     [3] Nguyen, T., Chen, Z., & Lee, J. (2020). Dataset meta-learning from kernel ridge-regression. arXiv preprint arXiv:2011.00050.

---

> > ### Comment · Reviewer_mfyC · 2023-08-19
> >
> > Thank you for taking the time to in-depth respond to my concern and for the additional evaluation. And I prefer to keep my score (5).

---

### Author Rebuttal · Authors · 2023-08-08

We deeply thank the reviewers for providing us with both positive feedback and valuable constructive criticism. Your professional review and careful reading have already helped us improve our work. We have thoroughly addressed all the comments raised during the initial review. If further clarity is required, please do not hesitate to reach out. Your engagement is highly valued.


We thank the reviewers for providing the following **Positive feedback**:

1. Rev BMB1: “The paper is the first attempt to theoretically guarantee the existence of dataset distillation”| Rev mfyC: fills a gap in the literature|  Rev piM1: “First theoretical work in dataset distillation + theoretical results are well established”| Rev uCrm: "The paper derives valid error bounds for KRR-based dataset distillation methods"

2.  Rev mfyC: Experimental study justifying the theoretical bounds: The paper includes both analytical proofs and empirical validation to support the theoretical results| Reviewer BMB1: “Experimental results are seemingly consistent with the theoretical results + the experimental protocol was clear".

3. The writing is logical and consistent (Rev uCrm).

4. The contributions of this paper and how this paper inherits previous results are very clear (Rev uCrm).

5. The use of random Fourier features (RFF) and kernel ridge regression (KRR) provides a solid theoretical foundation for the analysis (Rev mfyC).

**Clarity of our theoretical results**

Following your insightful comments, we revised the writing, by providing details before each theorem/claim/definition. We also added an intuition paragraph behind the proof idea.

Specifically, we added the following before Theorem 2:

The following theorem bounds the difference (additive approximation error) between (i) The MSE loss between the ground truth labels and the predictions obtained by applying Kernel Ridge regression (KRR) on the raw (original) data, and (ii) The MSE between the ground truth labels and the predictions obtained when applying Ridge regression on the mapped (full) training data via random Fourier features (RFF).

With this in mind, the goal of Theorem 2 is to set the minimal dimension of the RFF which yields the desired additive approximation ($4 \lambda$). The intuition, in our context, behind using this theorem, is to link the dimension of the RFF with the size of the distilled set in Theorem 3.

To that end, we use this error bound and sufficient size (of the minimal dimension of the RFF) to provide proof of the sufficient small size of the distilled set. This is done in Theorem 3. We added the following details to further explain it:

*The goal of Theorem 3.* is to prove the existence of a small distilled set $S$ (its size is a function of the minimal dimension of the RFF mapping required to ensure the provable additive approximation stated in Theorem 2) satisfying that:

   (i) The Ridge regression model trained on the mapped training data via RFF is identical to that of the Ridge regression model trained on the mapped small distilled set via RFF,

   (ii) more *importantly* there exists a KRR solution formulated for $S$ with respect to the loss of the whole big data $X$, which approximates the KRR solution on the whole data $X$ (which is the goal of KRR-based dataset distillation techniques). Thus,

   (iii) we derive bounds on the difference (approximation error) between (1) The MSE between the ground truth labels of the full data and their corresponding predictions obtained by the specific KRR model (we previously described) on our distilled set and (2) The MSE between the ground truth labels and the predictions obtained when applying KRR on the whole data $X$.

*Main idea.* The heart of our approach lies in connecting the minimal dimension of the RFF required for provable additive approximation and the size of the distilled set. This is first done by showing that the distilled set can be any set $S$ of instances from the input space (e.g., images) and their corresponding labels, *as long as the corresponding labels must maintain a certain property*. Specifically speaking, the labels of the distilled set need to be in correlation with the normal of the best hyperplane found to fit the mapped training data via RFF $\tilde{\mathbf{X}}$ via the Ridge regression model trained on $(\tilde{\mathbf{X}},y)$, i.e., $(\tilde{\mathbf{S}}^T\tilde{\mathbf{S}} +$ $\lambda n$ $s_\phi$ $\lambda$ $I_{s_\phi})$ $(\tilde{\mathbf{X}}^T$ $\tilde{\mathbf{X}}$ + $\lambda n s_\phi$ $\lambda I_{s_\phi}$ $)^{-1}$ $\tilde{\mathbf{X}}^T y$ $= \tilde{\mathbf{S}}^T y_{\mathbf{S}}$.

From here, the idea hinges upon showing the existence of a KRR model (represented by a prediction function) that would be dependent on the prediction function that can be obtained from applying the Ridge regression problem to the mapped full training data via RFF.

With such a model, the idea is to retrieve the predictions obtained when using the Ridge regression problem from the mapped training data via RFF via the use of some KRR model used on the distilled set.
We thus show that through careful mathematical derivations, equation reformulation (involving $\beta$), and solving a system of equations, one is able to show the existence of a KRR solution that would allow us to use Theorem 2. Finally, to obtain our bounds, we also rely on the use of the weak triangle inequality.
To that end, we now utilize the described KRR model on the distilled data together with Theorem 2 to achieve (iii).

*For Remark 6*, we note that it is an immediate result of Theorem 3. In Theorem 3, the approximation error is a function of a parameter $\tau$ that is related to which version of the weak triangle inequality is being used. Setting $\tau = 2$, we achieve the bound in Remark 6., which is a simplified version of Theorem 3.

Finally, for better readability and completeness, we have restated the proof of Theorem 2 from [LTOS21] in the appendix of the paper.
Thanks for the valuable comments.

---

### Author Response · Authors · 2023-08-11
**Looking forward to our discussion**

Dear Reviewers, Program Chairs, Area Chairs, and Senior Area Chairs,

Above all, we wish to express our sincere thanks and deep appreciation for dedicating your time to review and assess our work, along with offering us valuable insights. Your professional feedback has played a pivotal role in significantly improving our article.

We are delighted to update you that, due to your insightful comments, we have successfully tackled all the raised concerns within our paper and subsequently made the necessary updates, resulting in a much clearer manuscript (the modifications are given in the rebuttal section).

As the discussion period kicks off, we are enthusiastic about maintaining an ongoing conversation with all the reviewers. Please don't hesitate to bring up any concerns you may have, and we will be more than willing to promptly attend to them.

We express our gratitude to Reviewer uCrm for actively participating in the discussion and sharing their insights on our response. We firmly believe that such engagement is immensely valuable for enhancing the paper, and it's truly encouraging to witness the positive outcomes of the open review process.

Thank you very much.

---------------------------------------------------------------------------------------
**Main changes following your comments**

For your convenience, here is a summary of the big changes following your comments:

 1. Clarity of the paper: The reviewers raised concerns regarding the clarity of the theoretical results. Thus, we have revised the writing, and above all, we explained every theorem in our work, providing details about the idea and explanation regarding the derivations - you can find this update summarized in the rebuttal in the section “Clarity of our theoretical results” (https://openreview.net/forum?id=XWYv4BNShP&noteId=RA4SAIBzsG)
 2. Experimental results: We have added more experiments on the CIFAR dataset (attached) as requested by the reviewers, and as a bonus,  we are now running other results for SVHN. Finally, we are also adding more results for the multi-class case, which will be added before the discussion period ends.
 3. Provided more details regarding the size of the distilled set, how it is determined, when is it smaller, and when is it larger.

Finally, we have addressed all of the minor comments.

---

### Author Response · Authors · 2023-08-14
**Multiclass + CIFAR10 results**

In this comment, we share a link for the most recent experiments conducted following the reviewers' requests. In this **[link](https://postimg.cc/gallery/P8ksXYj)**, you can find new graphs for the multiclass case justifying our bounds and the CIFAR10 results.

Once we get results for SVHN, we will provide the results.

---

> ### Author Response · Authors · 2023-08-15
> **SVHN Results**
>
> In this comment, we share a link for the most recent experiment with respect to the SVHN dataset; please see this **[link](https://postimg.cc/gallery/P8ksXYj)**.

---

### Author Response · Authors · 2023-08-21
**Thank you + Summary**

Dear Reviewers, ACs, and SACs,

We would like to emphasize our thankfulness for the openly communicated review process, and to the reviewers who made this process very practical and beneficial by communicating with us, asking and responding to questions, suggesting many fruitful suggestions, and raising their scores following the helpful discussion - we truly appreciate that.

We are happy to note that we have improved our manuscript following your insightful review, by (i) adding all of the requested additional experimental results, (ii) adding full details and explanation about the theory (those details have also been provided here in the rebuttal and discussion) improving the readability of the paper, and (iii) providing more details regarding the size of the distilled set, how it is determined, when is it smaller, and when is it larger and why such a set exists.

With the conclusion of the discussion period drawing near, we want to reaffirm our availability to promptly address any remaining concerns or comments that may arise. At the same time, we hope we were able to adequately address all the reviewers’ concerns and issues encouraging you to accept our paper.

Thank you and we look forward to hearing back,

The authors

---

### Decision · Program_Chairs · 2023-09-21

**Decision:**

Accept (poster)

**Comment:**

The paper provides a theoretical analysis of dataset distillation. Utilizing the KRR approach, the authors provide the first formal proof regarding the existence of a small distilled dataset capable of effectively approximating the full input set. They also engage in discussions concerning the conditions and size specifications for such a distilled dataset.

Given that this marks the first theoretical analysis within the field of dataset distillation, all the reviewers, as well as the AC, acknowledge its significant contribution. However, the original submitted version of this paper lacks clarity in its presentation, leading to numerous questions from reviewers. After multiple rounds of communication, reviewers have been satisfied with the correctness of the analysis. Consequently, we recommend the paper for acceptance and encourage the authors to incorporate all the additional experiments and discussions in the camera version.